# Activated astrocytes attenuate neocortical seizures in rodent models through driving Na$^+$-K$^+$-ATPase

Junli Zhao[1,2,5], Jinyi Sun[2,5], Yang Zheng[1], Yanrong Zheng[1], Yuying Shao[2], Yulan Li[1], Fan Fei[1,2], Cenglin Xu [1], Xiuxiu Liu[2], Shuang Wang[3], Yeping Ruan[1], Jinggen Liu [1], Shumin Duan [4], Zhong Chen [1,2,3] ✉ & Yi Wang [1,2,3] ✉

Epileptic seizures are widely regarded to occur as a result of the excitation-inhibition imbalance from a neuro-centric view. Although astrocyte-neuron interactions are increasingly recognized in seizure, elementary questions about the causal role of astrocytes in seizure remain unanswered. Here we show that optogenetic activation of channelrhodopsin-2-expressing astrocytes effectively attenuates neocortical seizures in rodent models. This anti-seizure effect is independent from classical calcium signaling, and instead related to astrocytic Na$^+$-K$^+$-ATPase-mediated buffering K$^+$, which activity-dependently inhibits firing in highly active pyramidal neurons during seizure. Compared with inhibition of pyramidal neurons, astrocyte stimulation exhibits anti-seizure effects with several advantages, including a wider therapeutic window, large-space efficacy, and minimal side effects. Finally, optogenetic-driven astrocytic Na$^+$-K$^+$-ATPase shows promising therapeutic effects in a chronic focal cortical dysplasia epilepsy model. Together, we uncover a promising anti-seizure strategy with optogenetic control of astrocytic Na$^+$-K$^+$-ATPase activity, providing alternative ideas and a potential target for the treatment of intractable epilepsy.

Neocortical epilepsy is a relatively common type of epilepsy induced by various acquired factors, including brain trauma, infections, neoplasms, or vascular malformations[1]. Seizures in neocortical epilepsy arise from a focal neural network that is limited to a localization-related hemisphere and can easily develop resistance to current therapeutic interventions[2–4]. The central dogma explaining seizure genesis in epilepsy largely focuses on a neuro-centric theory of "excitability-inhibition" imbalance among neurons. Recently emerging evidence has also suggested a critical role for astrocytes in epilepsy, based on their multifaceted roles in buffering extracellular ions, clearing extracellular neurotransmitters, releasing gliotransmitters, and their

production of inflammatory mediators[5–8]. Although astrocyte-neuron interactions are known to be involved in seizure activity, it has been difficult to separate the precise impact of astrocytes from those of nearby neurons, and thus elementary questions about the role of astrocytes in the pathogenesis of seizure remain unresolved. One major hurdle has been the lack of a method to selectively modulate astrocyte activity. Optogenetics has recently emerged as one such method to selectively manipulate unidirectional astrocyte-to-neuron signaling[9,10], which has led to an improved understanding of how astrocytes can directly and actively modulate the excitability of neurons and neuronal networks[11–13]. However, the role of astrocytes in

[1]Key Laboratory of Neuropharmacology and Translational Medicine of Zhejiang Province, School of Pharmaceutical Sciences, Zhejiang Chinese Medical University, Hangzhou, China. [2]Institute of Pharmacology & Toxicology, College of Pharmaceutical Sciences, Zhejiang University, Hangzhou, China. [3]Epilepsy Center, Department of Neurology, Second Affiliated Hospital, School of Medicine, Zhejiang University, Hangzhou, China. [4]Key Laboratory of Medical Neurobiology of the Ministry of Health of China, School of Brain Science and Brain Medicine, Zhejiang University, Hangzhou, China. [5]These authors contributed equally: Junli Zhao, Jinyi Sun. ✉e-mail: chenzhong@zju.edu.cn; wang-yi@zju.edu.cn

cortical seizure in vivo remains poorly understood. In this study, we used optogenetic approaches to selectively manipulate astrocytes. Notably, we found that optogenetic activation of channelrhodopsin-2 (ChR2)-expressing astrocytes can effectively attenuate the severity of neocortical seizures and activity-dependently inhibit high-firing pyramidal neurons during cortical seizure. These effects did not rely on classical calcium signaling pathways in astrocytes, but instead were dependent on astrocytic Na⁺-K⁺-ATPase function.

## Results

### Optogenetic stimulation of ChR2-expressing astrocytes attenuates neocortical seizure

To selectively manipulate astrocytes, we injected a Cre-dependent channelrhodopsin-2 (ChR2)-EYFP AAV virus into the primary motor cortex (M1) of *GFAP-Cre* mice[11,14] (henceforth referred to as *GFAP-ChR2^M1* mice) (Fig. 1a). Immunostaining showed restricted ChR2-EYFP expression in the M1 region (Fig. 1b), and this was primarily co-localized with the astrocyte marker GFAP ($69.35 \pm 3.28\%$ of EYFP⁺ cells were GFAP⁺ astrocytes and $59.07 \pm 1.17\%$ of GFAP⁺ astrocytes expressed EYFP). Importantly, little overlap was observed with NeuN-labeled neurons ($1.28 \pm 0.25\%$ of EYFP⁺ cells were NeuN⁺ cells and $1.42 \pm 0.54\%$ of NeuN⁺ neurons expressed EYFP, Fig. 1c). ChR2-expressing astrocytes exhibited an inability to produce action potentials, but exhibited reliable activation of inward currents upon blue light stimulation (20 Hz, 473 nm) (Fig. 1d). To interrogate the role of M1 astrocytes in neocortical seizures in vivo, we used optogenetics to selectively stimulate astrocytes during neocortical seizures induced by administration of kainic acid (KA)[15], a potent agonist of the kainate receptors (Fig. 1e). Optogenetic stimulation (473 nm, 20 Hz, 5 mW, 10 ms, 30 s on/off duty cycle) of astrocytes substantially attenuated seizure progression (Fig. 1f) by both abating the seizure stage (Fig. 1g) and prolonging seizure onset (Fig. 1h). Importantly, optogenetic stimulation of astrocytes completely eliminated the occurrence of generalized seizures (GS) in 8/8 *GFAP-ChR2^M1* mice and prolonged the latency from $43.74 \pm 4.55$ min to the cut-off time of 90 min (Fig. 1i, j). The anti-seizure effects were similar in photostimulation of astrocytes with 1 Hz and 5 Hz blue light (Supplementary Fig. 1), indicating optogenetic stimulation of astrocytes is not frequency-dependent.

To investigate the anti-seizure time window of astrocyte stimulation in seizure development, we applied astrocyte stimulation in the early and late phases of neocortical seizures. We defined stage 2 as the demarcation point of the early and late phases. Interestingly, even early phase stimulation of astrocytes persistently attenuated the severity of seizures in all phases (Fig. 1f–j), including abating the seizure stage (Fig. 1g) and delaying the onset of seizures (Fig. 1h). In early phase stimulation, 87.5% (7/8) of mice remained GS-free and the latency to GS was $87.75 \pm 2.25$ min, suggesting that the anti-seizure effects produced by astrocyte stimulation can remain over a long duration. For late phase stimulation, the seizure severity was also reduced (Fig. 1f, g), although the onset of seizures was not significantly different from that of the control EYFP group (Fig. 1h). Notably, 62.5% (5/8) of mice in the ChR2-expressing group remained GS-free, with a significant increase in the latency to GS (of $84.60 \pm 2.90$ min; Fig. 1i, j), indicating optogenetic stimulation of astrocytes inhibited the spread of neocortical seizures. EEG spectra power were also significantly reduced and delayed in *GFAP-ChR2^M1* mice (Fig. 1k).

To further examine whether photostimulation of ChR2-expressing astrocytes in a remote brain region could suppress neocortical seizure, we introduced ChR2-EYFP virus into the left M1 (LM1) of *GFAP-Cre* mice to conduct photostimulation and injected KA into the right M1 (RM1) to induce seizures (Fig. 1l). Notably, photostimulation of astrocytes in the left M1 attenuated seizure stage progression, as determined by seizure stage (Fig. 1m, n), prolonged latency to GS (Fig. 1p), and number of GSs (Fig. 1q). No changes were observed in the mean duration to seizure onset, suggesting that

astrocyte stimulation in the distal cortex did not affect the initiation of seizure but inhibited the seizure spread (Fig. 1o). Overall, these data demonstrate that optogenetic stimulation of M1 astrocytes attenuates the severity of KA-induced neocortical seizures with the advantages of a wide therapeutic window and large-space efficacy.

We next assessed the ability of astrocyte stimulation to attenuate seizures evoked by intra-cortical administration of the chemo-convulsant pilocarpine[16] (Supplementary Fig. 2a). In this model, *GFAP-ChR2^M1* mice exhibited prolonged latency to GS (Supplementary Fig. 2b) and lower death rate (2/8 mice in *GFAP-ChR2^M1* group vs. 5/8 in *GFAP-EYFP^M1* group, Supplementary Fig. 2c). A comparison of the average frequency of epileptic spikes also revealed a pronounced inhibitory effect of astrocyte photostimulation (Supplementary Fig. 2d, e). Similarly, the EEG power was significantly reduced in *GFAP-ChR2^M1* mice (Supplementary Fig. 2d, f). Thus, optogenetic stimulation of astrocytes is also effective at attenuating pilocarpine-evoked cortical seizures.

### Optogenetic stimulation of astrocytes activity-dependently suppressed firing of cortical pyramidal neurons

Next, we aimed to determine whether photostimulation of astrocytes modulates neuronal activity. To this end, we utilized an in vivo single-unit recording system to record the firing activity of pyramidal neurons during photostimulation of astrocytes (Fig. 2a, b). In naive *GFAP-ChR2^M1* mice, photostimulation of M1 astrocytes altered the firing rate of pyramidal neurons in different modes (Fig. 2c). The firing of pyramidal neurons with low-frequency basal activity (0–1 Hz) following photostimulation of astrocytes was heterogeneous: 32.35% (11/34) of neurons gradually decreased, 29.41% (10/34) increased, and 38.24% (13/34) had no change. In pyramidal neurons with 1–5 Hz basal activity, 53.13% (17/32) of neurons exhibited decreased firing while 46.87% (15/32) of neurons showed no response. All pyramidal neurons (24/24) with high-frequency basal activity (>5 Hz) were inhibited upon photostimulation of astrocytes. The representative peri-event raster histograms and time-frequency statistical charts of pyramidal neurons showed 0–1 Hz neurons exhibited no obvious change, 1–5 Hz firing neurons were mildly inhibited and >5 Hz firing neurons were substantially inhibited by photostimulation of astrocytes (Fig. 2d–f). Delayed recovery was observed following inhibition of pyramidal neurons (Fig. 2e, f). These results suggest that optogenetic stimulation of astrocytes activity-dependently suppressed the firing rate of pyramidal neurons.

Generally, the firing rate of pyramidal neurons is relatively low under steady-state conditions, but high-frequency firing is observed in pre-seizure or seizures states[17,18]. Given our results in naïve mice indicating that neurons exhibiting the highest firing frequency were most susceptible to inhibition upon photostimulation of astrocytes, we next sought to record the firing activity of pyramidal neurons during blue light stimulation of astrocytes in KA-induced seizures in *GFAP-ChR2^M1* mice (Fig. 2g). Compared with interictal period recordings, the firing rate of pyramidal neurons was significantly increased (from $1.66 \pm 0.40$ Hz to $4.85 \pm 0.61$ Hz) during the seizure period (Fig. 2h). Following KA administration, 40% (4/10) of pyramidal neurons with 0–1 Hz firing rates were inhibited (50% (5/10) no response), 60% (6/10) of pyramidal neurons with 1–5 Hz firing rates were inhibited (40% (4/10) no response), and 100% (10/10) of pyramidal neurons with high-frequency firing (>5 Hz) were inhibited in response to photostimulation of astrocytes (Fig. 2i). The time-frequency statistical charts showed the firing rate of pyramidal neurons in seizure status were gradually inhibited and had no recovery (Fig. 2j). This phenomenon was in agreement with the long-term anti-seizure effects produced by astrocyte stimulation in KA-induced neocortical seizures.

We also recorded the firing activity of pyramidal neurons in RM1 during photostimulation in distal cortex LM1 (Fig. 2k). Similarly, the firing rate of pyramidal neurons with 0–5 Hz activity in response to

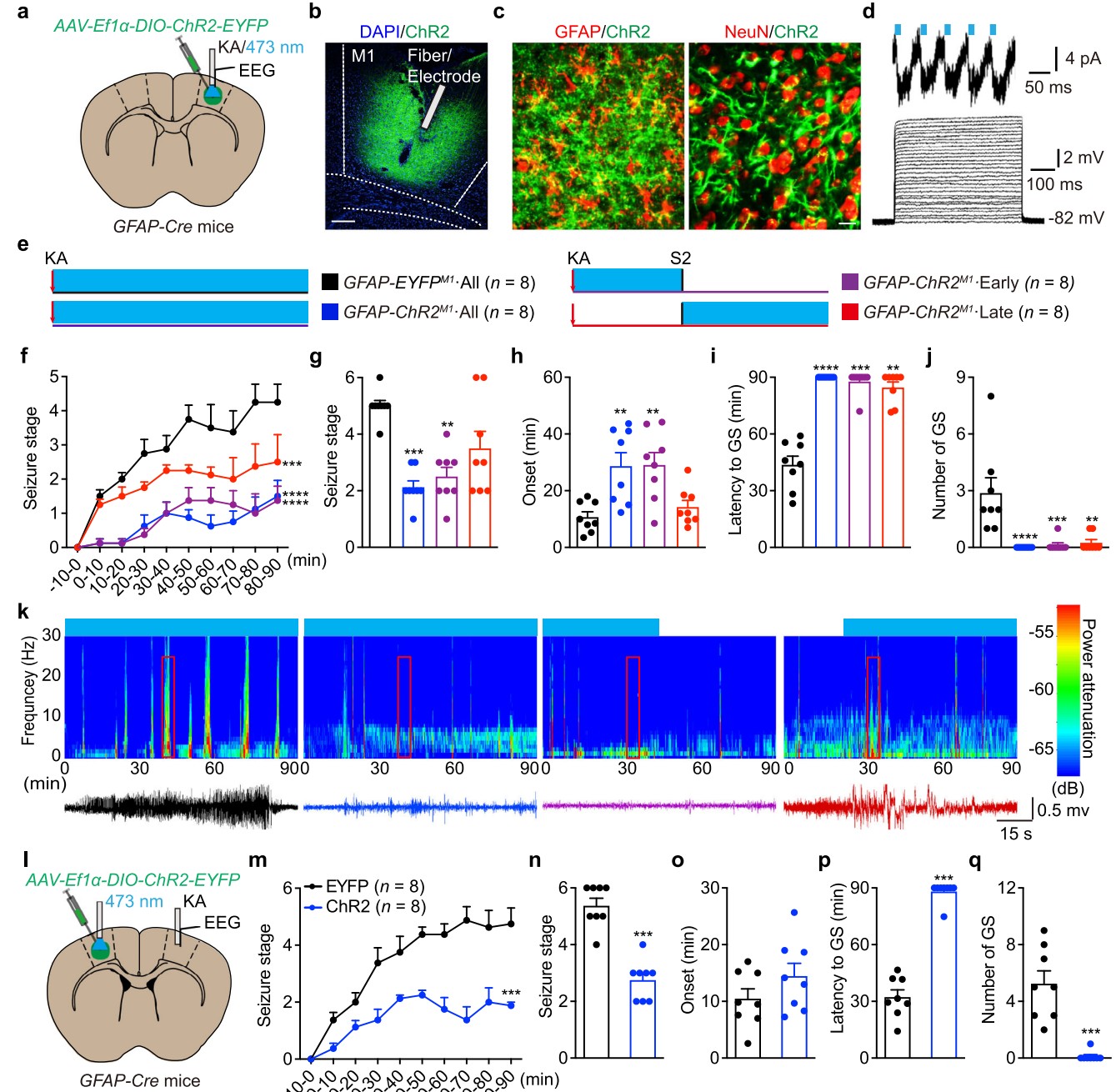

**Fig. 1 | Optogenetic stimulation of ChR2-expressing astrocytes attenuates KA-induced neocortical seizures. a** Schematic of viral injection, stimulation and EEG recording in *GFAP-ChR2^{M1}* mice. **b** Fluorescent image showed restricted expression of ChR2-EYFP in M1. Scale bar, 200 μm. **c** Left, fluorescent image of ChR2 (green) showed co-localization with GFAP⁺ astrocytes (red). Right, fluorescent image of ChR2 (green) showed no co-localization with NeuN⁺ neurons (red). Scale bar, 20 μm. **d** Light pulses induced inward currents in ChR2-expressing astrocyte in a cortical slice. Voltage responses from a ChR2-expressing astrocyte evoked by 10-pA step current from 0 to 300 pA. **e** Paradigms of blue light stimulation in different phases of KA-induced seizures. **f** Effects of optogenetic stimulation of astrocytes on the development of seizure stage during 90 min after KA injection. **g**–**j** Effects of different-phase optogenetic stimulation of astrocytes on seizure stage (**g**), EEG onset (**h**), latency to GS (**i**) and number of GSs (**j**) in KA-induced seizures. **k** Representative EEGs and corresponding energy spectra recorded from the M1 during KA-induced seizures. **l** Schematic of viral injection (LM1), stimulation (LM1) and EEG recording (RM1) in *GFAP-ChR2^{M1}* mice. **m** Effects of optogenetic stimulation of LM1 astrocytes on the development of seizure stage during 90 min after KA injection. **n**–**q** Effects of optogenetic stimulation of LM1 astrocytes on seizure stage (**n**), EEG onset (**o**), latency to GS (**p**) and number of GSs (**q**) after RM1 KA injection. $**p < 0.01$, $***p < 0.001$, $****p < 0.0001$ compared with EYFP control. Data shown as mean ± s.e.m. The number of mice used is indicated in figures. For detailed statistical information, see Supplementary Data 1. Source data are provided as a Source Data file.

photostimulation of astrocytes was heterogeneous (Fig. 2l, m). For 0−1 Hz firing pyramidal neurons: 33.33% (4/12) neurons were decreased, 16.67% (2/12) neurons were increased, and 50% (6/12) neurons had no change. For 1−5 Hz firing pyramidal neurons: 50% (4/8) neurons were decreased, 12.5% (1/8) neurons were increased, and 37.5%

(3/8) neurons had no change. Finally, high-frequency (>5 Hz) firing pyramidal neurons were 83.33% (5/6) inhibited and had delayed recovery (Fig. 2l, n). This phenomenon was in agreement with the anti-seizure effects observed in vivo after stimulation of astrocytes in remote cortical regions in KA-induced neocortical seizures.

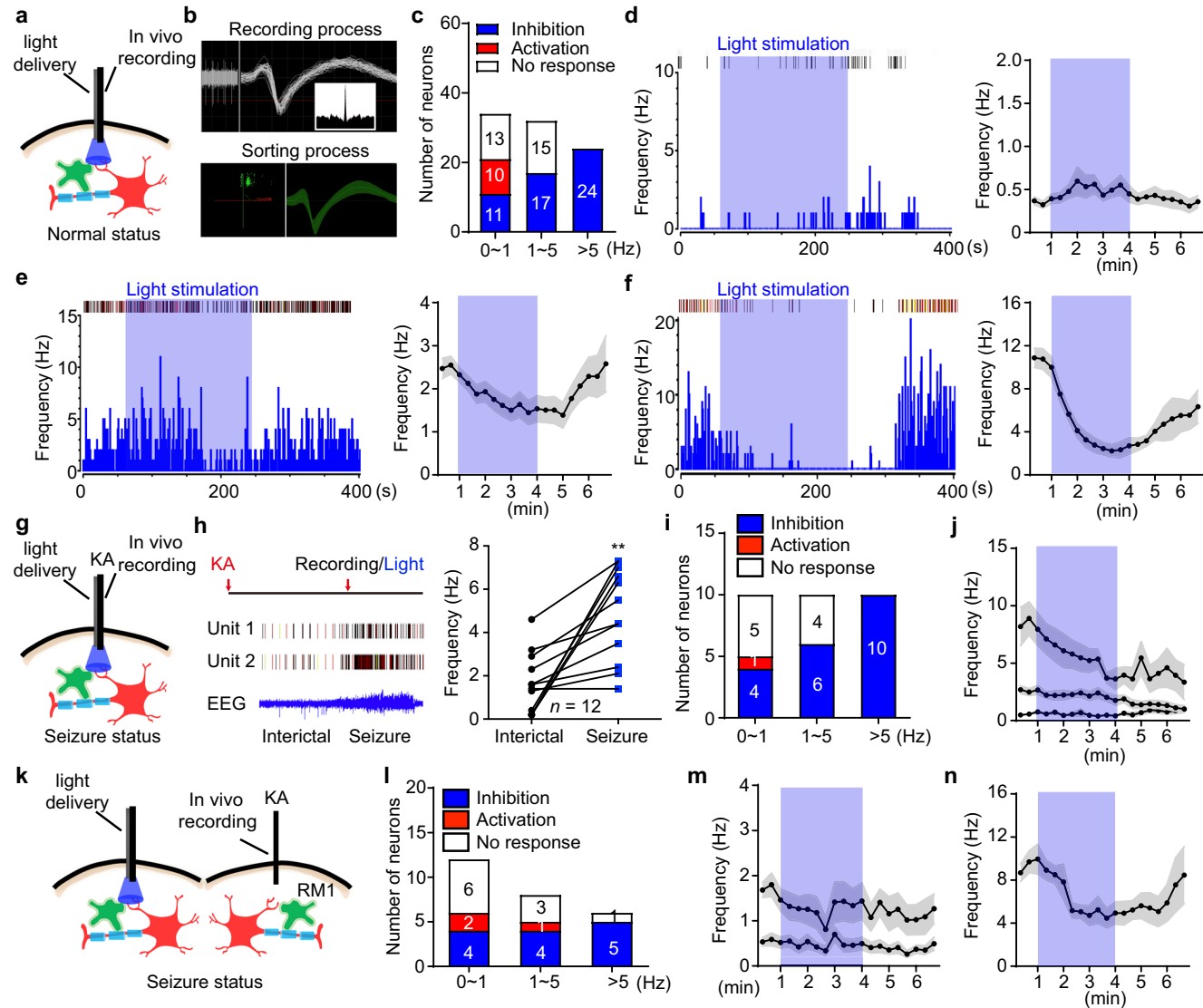

**Fig. 2 | Optogenetic stimulation of ChR2-expressing astrocytes activity-dependently inhibits firing of pyramidal neurons. a** Schematic of in vivo single-unit recording in naïve *GFAP-ChR2^MI* mice. Astrocyte (green), Neuron (red). **b** The online recording process and offline sorting process of in vivo single-unit recording of putative pyramidal neuron with wide waveform. Inset panel, corresponding protrude autocorrelation of a representative putative pyramidal neuron. **c** Summary data of the responses of recorded M1 pyramidal neurons during astrocyte stimulation. **d–f** Representative peri-event raster histograms and time-frequency statistical charts showed the response of blue light stimulation in different frequency groups. **g** Schematic of in vivo single-unit recording in *GFAP-ChR2^MI* mice in KA-induced seizure status. **h** Left panel, representative spikes and local field potentials of M1 pyramidal neurons during interictal and seizure phases. Right panel, quantification of the frequency of M1 pyramidal neurons during interictal and seizure phases. **i** Summary data of the responses of recorded M1 pyramidal neurons during astrocyte stimulation in KA-induced seizure status. **j** Time-frequency statistical charts showed the response of blue-light stimulation in different frequency groups in KA-induced seizure status. **k** Schematic of in vivo single-unit recording in RM1 after RM1 KA injection and LM1 blue light stimulation. **l** Summary data of the responses of recorded RM1 pyramidal neurons during astrocytes stimulation after RM1 KA injection and LM1 blue light stimulation. **m**, **n** Time-frequency statistical charts showed the response of blue-light stimulation in different frequency groups in KA-induced seizure status. **\*\***$p < 0.01$ compared with interictal phase. Data shown as mean ± s.e.m. The number of mice used is indicated in figures. For detailed statistical information, see Supplementary Data 1. Source data are provided as a Source Data file.

Collectively, these data indicate that optogenetic stimulation of astrocytes could activity-dependently suppress the firing rate of pyramidal neurons.

**Optogenetic inhibition of pyramidal neurons alleviates KA-induced neocortical seizure with narrow therapeutic window**

Next, we tested whether optogenetic inhibition of pyramidal neurons in the cortex could similarly attenuate neocortical seizures. We injected a Cre-dependent ArchT-tdTomato AAV virus into the M1 of *CaMKII-Cre* mice (henceforth referred to as *CAMKII-ARCH^MI* mice). Photo-activation of the outward proton pump ArchT-tdTomato generally inhibited the firing rate of pyramidal neurons (Fig. 3a, b). We used similar photo-inhibition parameters (590 nm, 5 mW, constant, 30 s on/off duty cycle) and modes (all phase, early phase and late phase) in this experiment as were used in our astrocyte photostimulation experiments (Fig. 3c). The progression of seizure stage in the all phase photo-inhibition group was significantly attenuated (Fig. 3d, e). A trend toward prolonged seizure onset was observed in photo-inhibited mice (Fig. 3f). The GS-free percent was 87.5% (7/8) and the latency to GS was prolonged from 34.55 ± 5.91 min to 87.08 ± 2.92 min (Fig. 3g, h). Notably, neither early phase nor late phase photo-inhibition attenuated the progression of seizure stage (Fig. 3d). In early phase

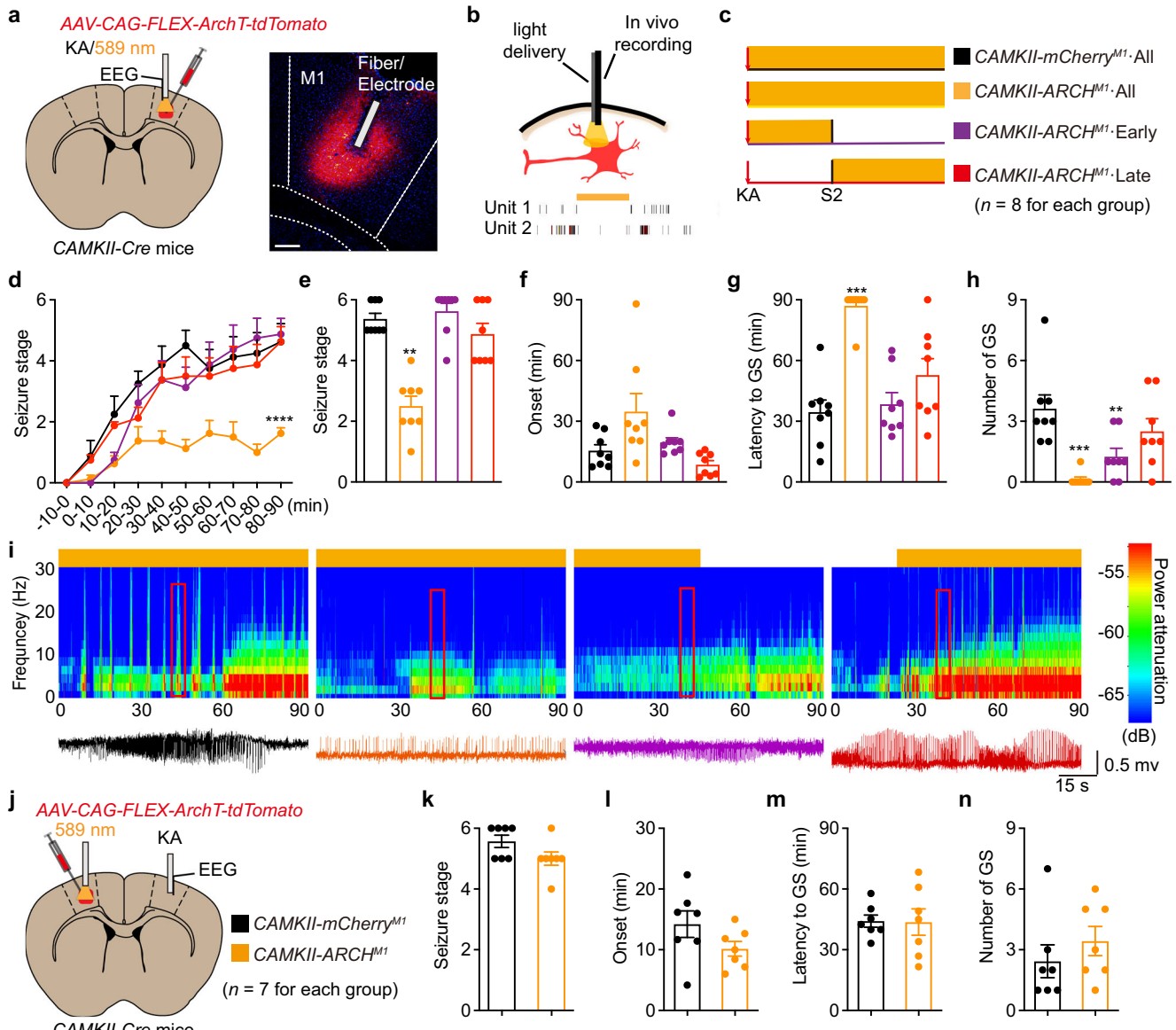

**Fig. 3 | Optogenetic inhibition of pyramidal neurons attenuates neocortical seizures with narrow therapeutic window. a** Left panel, schematic of viral injection, stimulation and EEG recording in RM1 of *CAMKII-ARCH*[M1] mice. Right panel, fluorescent image showed restricted expression of ArchT-tdTomato in M1. Scale bar, 200 μm. **b** Representative spikes of M1 pyramidal neurons showing the reduced firing rate in response to yellow light stimulation. **c** Paradigms of yellow light stimulation in different phases in KA-induced seizures. **d** Effects of optogenetic inhibition of RM1 pyramidal neurons on the development of seizure stage during 90 min after KA injection. **e–h** Effects of different-phase optogenetic inhibition of RM1 pyramidal neurons on seizure stage (**e**), EEG onset (**f**), latency to GS

(**g**) and number of GSs (**h**) in KA-induced seizures. **i** Representative EEGs and corresponding energy spectra of pyramidal neurons inhibition during seizure activity. **j** Schematic of viral injection (LM1), stimulation (LM1) and EEG recording (RM1) in *CAMKII-ARCH*[M1] mice. **k–n** Effects of optogenetic inhibition of LM1 pyramidal neurons on seizure stage (**k**), EEG onset (**l**), latency to GS (**m**) and number of GSs (**n**) after KA injection. **p < 0.01, ***p < 0.001 and ****p < 0.001 compared with mCherry control group. Data shown as mean ± s.e.m. The number of mice used is indicated in figures. For detailed statistical information, see Supplementary Data 1. Source data are provided as a Source Data file.

photo-inhibition, only the number of GSs was significantly decreased compared to the control group, but no long-term anti-seizure effects were observed in this condition (Fig. 3h, i). In addition, latency to GS in *GFAP-ChR2*[M1] mice was significantly longer than in *CAMKII-ARCH*[M1] mice (63.83 ± 19.33 min vs. 14.06 ± 5.57 min) after photostimulation withdrawal (Supplementary Fig. 3a, b). Meanwhile, 50% (5/10) of *GFAP-ChR2*[M1] mice were still GS-free even after photostimulation withdrawal for 120 min (Supplementary Fig. 3c, d). These results indicate that optogenetic inhibition of pyramidal neurons has a relatively narrow window to produce anti-seizure effects.

To further test the effect of optogenetic inhibition of pyramidal neurons on seizure spread, we conducted photo-inhibition in LM1 and

induced seizures in RM1 using KA in *CAMKII-ARCH*[M1] mice (Fig. 3j). Photo-inhibition did not produce anti-seizure effects, as no obvious change in seizure stage, seizure onset, latency to GS, or number of GSs was observed compared to control mice (Fig. 3k–n). Thus, optogenetic inhibition of pyramidal neurons in distal cortex did not inhibit seizure spread. Meanwhile, to test whether optogenetic modulation of pyramidal neurons or astrocytes in the M1 has a side effect on motor function, we compared locomotor function via open field test in *GFAP-ChR2*[M1] mice and *CAMKII-ARCH*[M1] mice. We found that optogenetic inhibition of pyramidal neurons in *CAMKII-ARCH*[M1] mice significantly decreased the total distance traveled during yellow light stimulation, whereas optogenetic stimulation of astrocytes in *GFAP-ChR2*[M1] mice

did not alter locomotor activity during blue light stimulation (Supplementary Fig. 4). All mice were self-controlled with the other light stimulation (Supplementary Fig. 4a, b). We therefore conclude that direct optogenetic inhibition of pyramidal neurons has only a narrow window in which it can produce anti-seizure effects, and impair motor function.

## Astrocytic calcium signaling is not required to produce the anti-seizure effects induced by optogenetic stimulation of astrocytes

ChR2 is a light-gated cation-selective membrane channel that mainly permits entry of $Na^+$ and $Ca^{2+}$ ions into cells[19]. Astrocytic calcium signaling plays a critical role in gliotransmitter release and modulation of neuronal activity[6]. We thus hypothesized that $Ca^{2+}$ infusion into astrocytes during ChR2 channel activation would release inhibitory gliotransmitters, thereby inhibiting neuronal activity and seizure intensity. To test this hypothesis, we used two different types of pharmacological blockers to inhibit astrocytic calcium signaling upon optogenetic stimulation, as ChR2-induced cytosolic $Ca^{2+}$ elevation in astrocytes is caused by either $Ca^{2+}$ release from intracellular $Ca^{2+}$ stores or extracellular sources[20,21]. One is a non-competitive inhibitor of the sarco/endoplasmic reticulum $Ca^{2+}$ ATPase, thapsigargin (TG), and the other is a cell-permeant $Ca^{2+}$ chelator, BAPTA-AM. To verify the functional inhibition of these two drugs, we injected a Cre-dependent GCaMP6m virus into GFAP-Cre mice and recorded calcium signaling during seizures (Fig. 4a). Astrocytic calcium signaling was increased during seizure and could be blocked by application of either TG or BAPTA-AM (Fig. 4b). Surprisingly, injection of TG or BAPTA-AM into the M1 in GFAP-ChR2$^{M1}$ mice revealed that neither of these $Ca^{2+}$ blockers could reverse the anti-seizure effect of astrocyte stimulation (Fig. 4c–g), suggesting that astrocytic calcium signaling is not required to produce the anti-seizure effects caused by optogenetic stimulation of astrocytes.

Next, we further tested whether increased astrocytic calcium signaling contributes to seizure pathogenesis using hM3Dq-mediated chemogenetic activation of astrocytes' intracellular $Ca^{2+}$ [22]. We injected both Cre-dependent GCaMP6m and hM3Dq viruses into GFAP-Cre mice and recorded calcium signaling in astrocytes (Fig. 4h). Astrocytic intracellular $Ca^{2+}$ was increased after CNO injection (Fig. 4i). However, hM3Dq-mediated enhancement of intracellular $Ca^{2+}$ did not have any anti-seizure effects (Fig. 4j–n). On the contrary, the latency to seizure onset was significantly reduced (Fig. 4l), which may be consistent with previous findings indicating that excitatory glutamate gliotransmission ($Ca^{2+}$-dependent) promotes seizure initiation[23–25]. Taken together, these results demonstrated that astrocytic calcium signaling indeed is involved in seizure initiation, but is not required for the anti-seizure effect of optogenetic stimulation of astrocytes.

## Astrocytic $Na^+$-$K^+$-ATPase mediates the anti-seizure effect of optogenetic stimulation of astrocytes

Beyond $Ca^{2+}$ influx, ChR2 activation also permits $Na^+$ entry into cells upon stimulation. We posited that $Na^+$ influx may activate the astrocytic $Na^+$-$K^+$-ATPase, enabling uptake of excessive extracellular $K^+$, which occurs during seizure states[26,27]. To test whether astrocytic $Na^+$-$K^+$-ATPase mediates the anti-seizure effect of astrocyte stimulation, we used a $Na^+$-$K^+$-ATPase inhibitor, ouabain[28], to block the function of $Na^+$-$K^+$-ATPase[29,30]. We found that intra-cortical injection of ouabain partially reversed the anti-seizure effects of optogenetic astrocyte stimulation, manifested by accelerated seizure stage development (Fig. 5c, d), shortened onset of seizures (Fig. 5e) and latency to GS (Fig. 5f) in comparison to vehicle injected mice. The percentage of GS-free mice decreased from 100% (8/8) to 12.5% (1/8) and the number of GSs were largely increased (Fig. 5g). These results suggest the involvement of astrocytic $Na^+$-$K^+$-ATPase in the anti-seizure effects of optogenetic stimulation of astrocytes.

To further verify the involvement of the astrocytic $Na^+$-$K^+$-ATPase, we designed an AAV in which the GFAP promotor drives expression of an shRNA targeting Atp1α2, the $Na^+$-$K^+$-ATPase α2 subunit, thus permitting astrocyte-selective knockdown of $Na^+$-$K^+$-ATPase function (Fig. 5a). We confirmed astrocytic-selective knockdown of $Na^+$-$K^+$-ATPase both in vitro and in vivo. The in vitro primary cortical astrocytes transfected with AAV-GFAP-shRNA(Atp1α2)-GFP showed no ATP1A2 protein expression in GFAP$^+$ astrocytes (Supplementary Fig. 5a). The in vivo cortical tissue injected with negative control AAV-GFAP-shRNA(NC)-GFP showed ATP1A2 protein co-localization with GFAP$^+$shRNA(NC)$^+$ astrocytes while cortical tissue injected with AAV-GFAP-shRNA(Atp1α2)-GFP showed significantly reduced ATP1A2 protein expression in GFAP$^+$shRNA(Atp1α2)$^+$ astrocytes (Fig. 5b). The shRNA(Atp1α2)-GFP was not co-localized with NeuN$^+$ cells (Supplementary Fig. 5b), further confirming the astrocytic-selective knockdown of the $Na^+$-$K^+$-ATPase α2 subunit. Knockdown of the astrocytic $Na^+$-$K^+$-ATPase α2 subunit completely reversed the anti-seizure effects produced by optogenetic stimulation of astrocytes, including the rate of seizure stage development, seizure onset, latency to GS and number of GSs (Fig. 5c–g). Meanwhile, we also observed that the astrocyte stimulation-induced inhibitory effects on pyramidal neuron firing rate were reversed by ouabain-treatment and shRNA-knockdown in GFAP-ChR2$^{M1}$ mice (Fig. 5h–k).

Given that the $Na^+$-$K^+$-ATPase is critically involved in buffering extracellular $K^+$, we next sought to test whether these functions are related to modulation of $K^+$ concentration in the extracellular space. First, we measured resting membrane potential (RMP) of astrocytes as an indirect reflection of $K^+$ equilibrium potential. We found that optogenetic stimulation of astrocytes increased RMP during light stimulation, but decreased RMP after light withdrawal, in both naïve and seizure status (Fig. 5l). Notably, in KA-induced seizure status, RMP was significantly decreased after optogenetic stimulation of astrocytes (Fig. 5m), suggesting that extracellular $K^+$ may be effectively buffered. Secondly, we directly measured extracellular $K^+$ concentration using microdialysis combined with plasma atomic emission spectroscopy[31]. We found that KA-induced seizure led to an increase in extracellular $K^+$ concentration, and that optogenetic stimulation of astrocytes rescued the seizure-induced elevation in extracellular $K^+$ concentration (Fig. 5n). It should be noted, ChR2 may also be permeable to $K^+$ (which would allow for $K^+$ efflux). Octeau et al. reported that ChR2 (H134R) activation in astrocytes and neurons caused a transient increase in extracellular $K^+$ to ~7.4 mM and ~8.8 mM, respectively, under physical condition[32]. However, in seizure status, the extracellular [K+] can rise to as high as 10–12 mM[33], which would likely limit $K^+$ release into the extracellular space during ChR2 activation. Our data showed that optogenetic activation of astrocytes largely decreases extracellular $K^+$ in seizure state.

Since astrocytic Kir4.1 is known to be important for extracellular $K^+$ buffering in seizures[34], we tested whether Kir4.1 contributed to the removal of extracellular $K^+$, and thereby the anti-seizure effects, induced by optogenetic stimulation of astrocytes. Firstly, we examined the expression of Kir4.1 in astrocytes via immunohistochemistry and found that Kir4.1 was primarily stained in cells with the characteristic stellate-shape of astrocytes, and was highly co-localized with the astrocyte marker GFAP, in both naïve and seizure mice (Supplementary Fig. 6a). Then, we applied the Kir4.1 blocker (VU0134992)[35] before KA-induced seizures with photostimulation of astrocytes (Supplementary Fig. 6b). Compared to vehicle, Kir4.1 blocker alone increased the number of GSs in control mice (Supplementary Fig. 6f), indicating that blocking Kir4.1 aggravates seizure severity in control mice. However, blocking astrocytic Kir4.1 could not reverse the anti-seizure effects produced by optogenetic stimulation of astrocytes, including the rate of seizure stage development, latency to GS and number of GSs (Supplementary Fig. 6c–f). These data indicated that the astrocytic

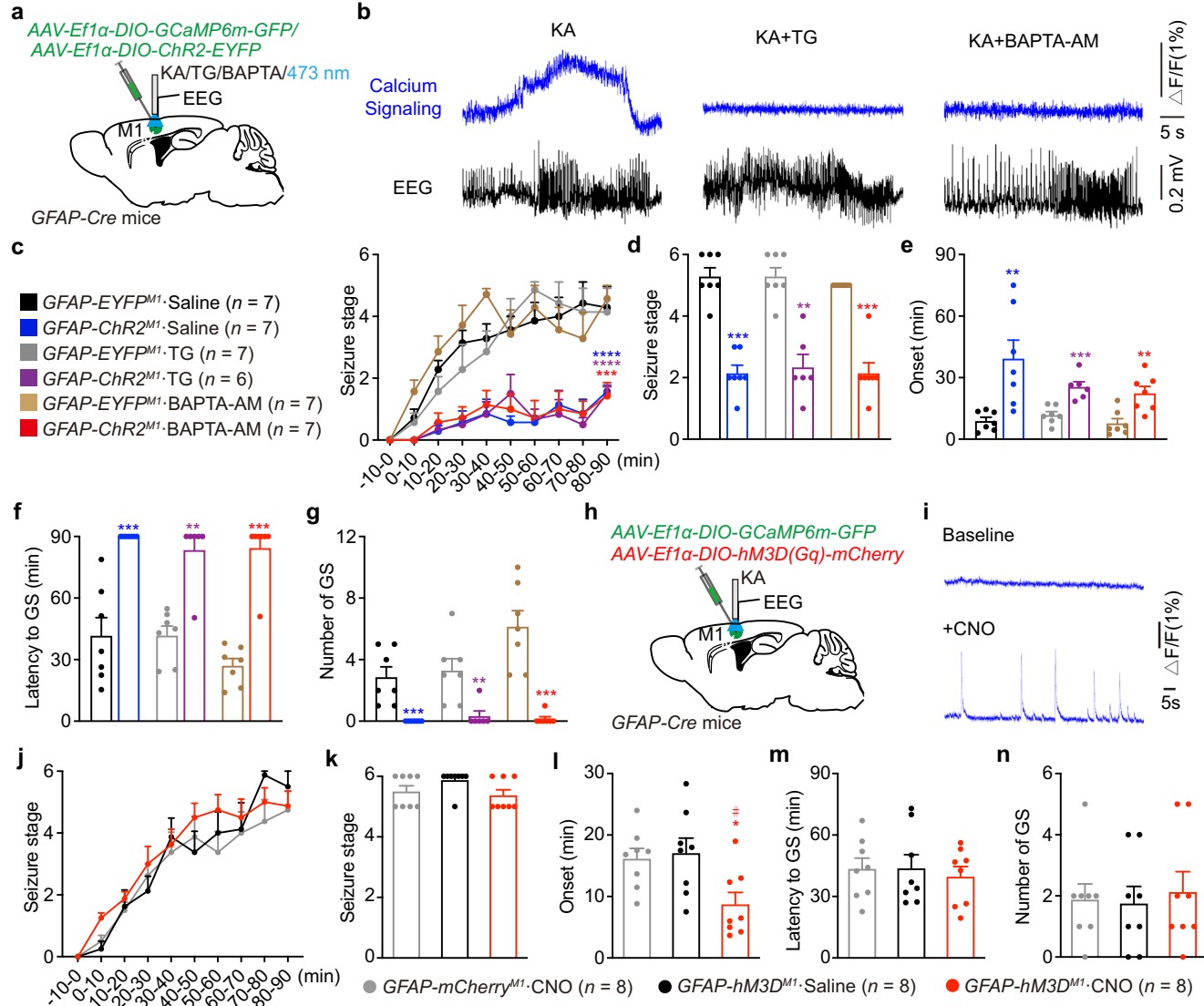

**Fig. 4 | Astrocytic calcium signaling is not required to produce the anti-seizure effects induced by optogenetic stimulation of astrocytes. a** *GFAP-Cre* mice were injected with Cre-dependent virus expressing GCAMP6m-GFP or ChR2-EYFP in the M1 and implanted with cannula and electrode for drug injection, blue light stimulation and EEG recording. **b** Typical calcium signals and corresponding seizure EEGs in KA or KA + TG/BAPTA-AM injection groups. Δ*F/F* represents the change in signal amplitude from the base level. **c** Effects of optogenetic stimulation of astrocytes on the development of seizure stage, in the condition of calcium signaling inhibition by TG or BAPTA-AM injection. **d–g** Effects of optogenetic stimulation of astrocytes on seizure stage (**d**), EEG onset (**e**), latency to GS (**f**) and number of GSs (**g**), in the condition of calcium signaling inhibition by TG or BAPTA-AM injection. **h** *GFAP-Cre* mice were injected with Cre-dependent virus expressing GCAMP6m-GFP and/or

hM3D(Gq)-mCherry in the M1 and implanted with cannula and electrode for KA injection and EEG recording. **i** Typical calcium signals before and after CNO injection indicating hM3Dq-mediated increase of calcium signaling. Δ*F/F* represents the change in signal amplitude from the base level. **j** Effects of chemogenetic activation of astrocytes on the development of seizure stage during 90 min after KA injection. **k–n** Effects of chemogenetic activation of astrocytes on the seizure stage (**k**), EEG onset (**l**), latency to GS (**m**) and number of GSs (**n**) in different groups. **p < 0.01, ***p < 0.001, ****p < 0.0001 compared with each control group. **l** *p < 0.05 compared with mCherry-CNO group, #p < 0.05 compared with hM3Dq-Saline group. Data shown as mean ± s.e.m. The number of mice used is indicated in figures. For detailed statistical information, see Supplementary Data 1. Source data are provided as a Source Data file.

$Na^+-K^+$-ATPase, but not Kir4.1, is required for the anti-seizure effects induced by optogenetic stimulation of astrocytes.

In addition, we found that astrocytic gap junctions are important for the large-space efficacy in anti-seizure effects[36], as the gap junction inhibitor carbenoxolone (CBX) reversed anti-seizure effects and the inhibitory effects on pyramidal neuronal firing rate produced by optogenetic stimulation of astrocytes both in local and remote cortical regions (Supplementary Fig. 7). Taken together, our results demonstrate that the astrocytic $Na^+-K^+$-ATPase-mediated buffering of $K^+$ contributes to activity-dependent inhibition of hyperexcitable pyramidal neurons during seizure and can attenuate neocortical seizures.

**Optogenetic stimulation of astrocytes rescues the seizure susceptibility of FCD rats via astrocytic $Na^+-K^+$-ATPase**

Finally, to test whether optogenetic-driven astrocytic $Na^+-K^+$-ATPase would have a therapeutic role in a chronic neocortical epilepsy model, we established the focal cortical dysplasia (FCD) model[37,38] (FCD is an important cause of intractable neocortical epilepsy) in embryonic day 18 (E18) rats, and tested their seizure susceptibility in the later adult period by i.p. injection of pentylenetetrazol (PTZ) (Fig. 6a). Loss of lamination pattern and neuronal depletion of the cortex were visible in the dysplastic rats, compared with nondysplastic rats (Fig. 6b). We also recorded cortical EEG from dysplastic rats which showed several forms

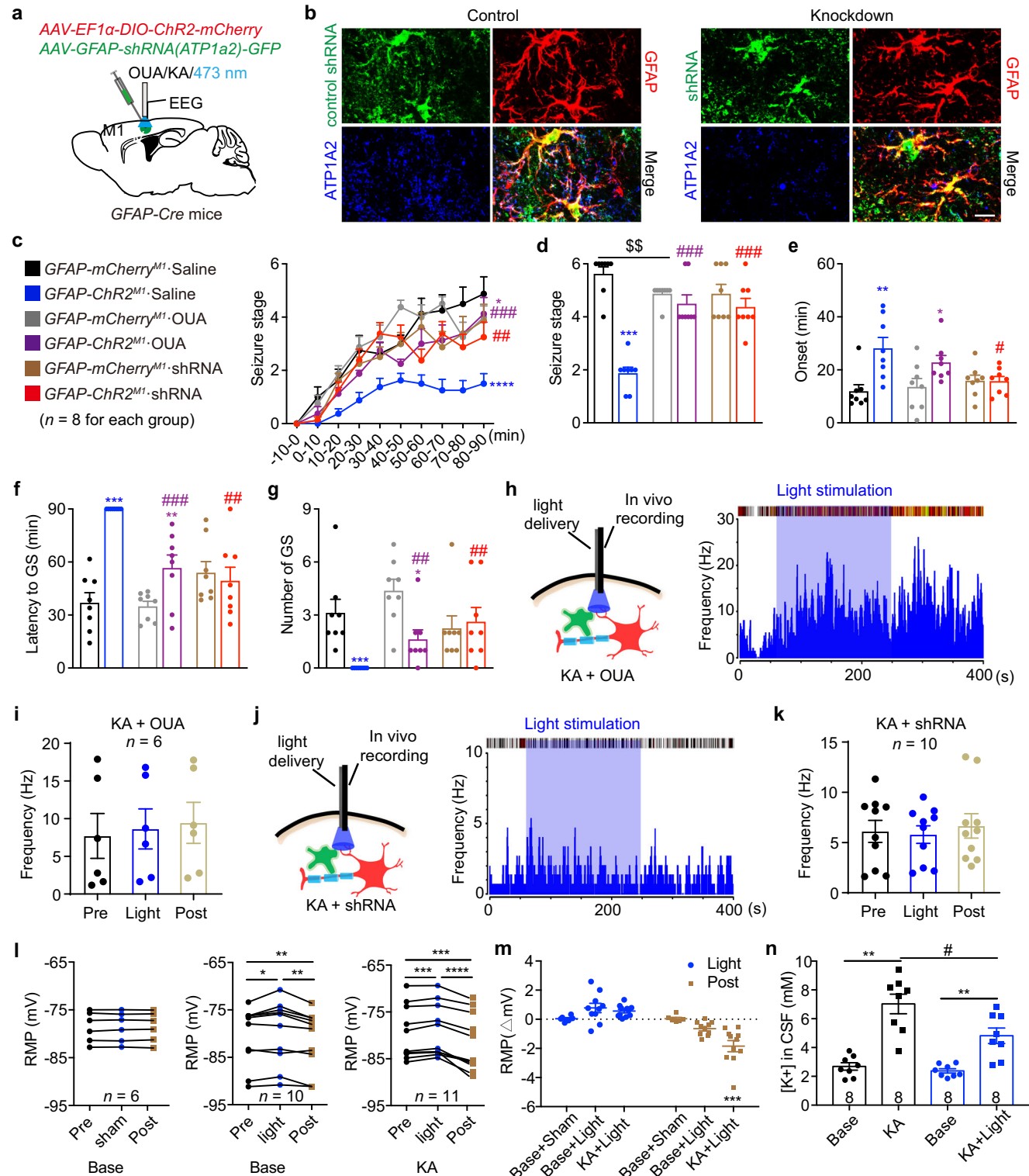

of spontaneous epileptic waves, including polyspikes, paroxysmal fast activity and rhythmic discharges (Fig. 6c). However, these abnormal epileptiform activities were occurred with extremlly low-frequency (~4 times/8 h), which make it impractical to test the effects of optogenetic stimulation of ChR2-expressing astrocytes on spontaneous seizures, as long-term (several hours) stimulating astrocytes with light would easily heat the tissue and thus induce cellular damage. This result at least indicates that FCD rats may increase neural excitability and capable of generating abnormal electrographic activity. To further reliably determine the seizure susceptibility of FCD rats, we injected

60 mg/kg PTZ into both control and FCD rats and found the increased seizure susceptibility in FCD rats (Fig. 6d–h), including increased seizure stage (Fig. 6e), shortened the latency to GS (Fig. 6g) and prolonged GS duration (Fig. 6h) in FCD rats, compared with control rats. We also tested Na$^+$-K$^+$-ATPase α2 protein expression via both immunohistochemistry and western blot in control and FCD rats. Representative confocal images showed that Na$^+$-K$^+$-ATPase α2 protein (ATP1A2) was expressed in astrocytic membranes and identified that FCD tissues tended to have a decreased membrane fraction of Na$^+$-K$^+$-ATPase α2 protein (Fig. 6i). Western blot results also showed a

**Fig. 5 | Astrocytic Na⁺-K⁺-ATPase mediates the anti-seizure effect of optogenetic stimulation of astrocytes. a** *GFAP-Cre* mice were injected with Cre-dependent ChR2-mCherry and/or pAAV-shortGFAP-MCS-EGFP-3FLAG-mir30shRNA(Atp1α2) in the M1 and implanted with cannula and electrode for KA injection, blue light stimulation and EEG recording. **b** Fluorescent images showed that shRNA (green) showed co-localization with GFAP⁺ astrocytes (red), and ATP1A2 protein co-localization with GFAP⁺shRNA(NC)⁺ astrocytes from control shRNA(NC) injected cortical tissue. ATP1A2 protein expression in the GFAP⁺ astrocytes is significantly reduced in shRNA(Atp1α2) injected cortical tissue. Scale bar, 20 μm. **c** Effects of optogenetic stimulation of astrocytes on the development of seizure stage, in the condition of astrocytic Na⁺-K⁺-ATPase inhibition. **d–g** Effects of optogenetic stimulation of astrocytes on seizure stage (**d**), EEG onset (**e**), latency to GS (**f**) and

number of GSs (**g**), in the condition of astrocytic Na⁺-K⁺-ATPase inhibition. **h–j** Representative peri-event raster histograms and statistical data of firing rate of pyramidal neurons in *GFAP-ChR2ᴹᴵ* mice in response to blue light stimulation during in vivo single-unit recording, in the condition of Na⁺-K⁺-ATPase inhibition by ouabain (**h, i**) or shRNA-knockdown (**j, k**). **l, m** Rest membrane potential of astrocytes (**l**) and their changes (**m**) before and after blue light stimulation in *GFAP-ChR2ᴹᴵ* slices under normal and KA incubation conditions. **n** In vivo CSF potassium concentration before and after KA incubation in *GFAP-ChR2ᴹᴵ* mice under sham and blue light stimulation conditions. \*$p < 0.05$, \*\*$p < 0.01$, \*\*\*$p < 0.001$, \*\*\*\*$p < 0.0001$; #$p < 0.05$, ##$p < 0.01$, ###$p < 0.001$. Data shown as mean ± s.e.m. The number of mice used is indicated in figures. For detailed statistical information, see Supplementary Data 1. Source data are provided as a Source Data file.

tendency of decreased expression of Na⁺-K⁺-ATPase α2 protein in FCD rats (Fig. 6j).

Next, we injected an AAV-GFAP-ChR2 virus into the M1 of FCD rat (henceforth referred to as *FCD-ChR2ᴹᴵ* rat) (Fig. 6k). To interrogate the role of M1 astrocytes in the FCD model in vivo, we used optogenetics to selectively stimulate astrocytes during neocortical seizures induced by administration of PTZ. Optogenetic stimulation (473 nm, 20 Hz, 5 mW, 10 ms, 30 s on/off duty cycle) of astrocytes substantially attenuated seizure progression in *FCD-ChR2ᴹᴵ* rats (Fig. 6m–p). Importantly, optogenetic stimulation of astrocytes significantly decreased the occurrence of GS in 8/10 *FCD-ChR2ᴹᴵ* rats and prolonged the latency from 531 ± 213.4 s to 1476 ± 216.4 s (Fig. 6o) as well as shortened the GS duration from 47.5 ± 14.64 s to 3.8 ± 2.641 s (Fig. 6p). These results indicates that optogenetic stimulation of astrocytes attenuates the enhanced seizure susceptibility in FCD rats.

To further verify the involvement of the astrocytic Na⁺-K⁺-ATPase in the anti-seizure effects of astrocyte stimulation in FCD rats, we injected *FCD-ChR2ᴹᴵ* rats with a Na⁺-K⁺-ATPase α2 shRNA driven by the GFAP promoter to selectively knockdown astrocytic Na⁺-K⁺-ATPase expression (Fig. 6k, l). Knockdown of astrocytic Na⁺-K⁺-ATPase α2 subunit reversed the anti-seizure effects produced by optogenetic stimulation of astrocytes in *FCD-ChR2ᴹᴵ* rats (Fig. 6m–p), including the seizure stage (Fig. 6m), latency to GS (Fig. 6o) and GS duration (Fig. 6p). We also injected Na⁺-K⁺-ATPase α2 shRNA into control rats to test the effect of astrocyte-selective knockdown of Na⁺-K⁺-ATPase function in control rats (Supplementary Fig. 8a). Knockdown of astrocytic Na⁺-K⁺-ATPase α2 subunit was found to aggravate the severity of seizures induced by administration of 60 mg/kg PTZ in control rats (Supplementary Fig. 8b–e), indicating that astrocytic Na⁺-K⁺-ATPase α2 is essential for the seizure condition. These results further demonstrated that astrocytic Na⁺-K⁺-ATPase was required for the anti-seizure effects induced by optogenetic stimulation of astrocytes in the FCD model. Taken together, optogenetic stimulation of ChR2-expressing astrocytes rescues the seizure susceptibility of FCD rats and could serve as a promising therapeutic strategy for intractable epilepsy with FCD.

## Discussion

Epilepsy represents a wide range of pathological neural network alterations characterized by recurrent episodes of excessive brain activity[39–41]. Thus, identifying the properties of seizure-producing networks may enable more efficient seizure control[42]. Although astrocyte-neuron interactions are known to be involved in seizure activity[43–45], it has proven difficult to separate the precise impact of astrocytes from that of nearby neurons in the causal underpinnings of seizure due to the lack of a specific approach to selectively modulate astrocytes. Here, by using optogenetics to selectively manipulate unidirectional astrocyte-neuron signaling, we found that photostimulation of ChR2-expressing astrocytes can effectively attenuate neocortical seizures. In the KA-induced seizure model, astrocyte

stimulation attenuated the progression of seizure stage, and perhaps most notably, completely eliminated the occurrence of GS. These anti-seizure effects can be effectively achieved in a wide frequency-independent window, as 1-, 5-, and 20-Hz stimulation all work. Further, astrocyte stimulation can produce anti-seizure effects at both early and late phases, and it can effectively delay the occurrence of GS even in remote cortical regions, suggesting it can achieve efficient control of seizure initiation and spread. We also verified the anti-seizure effects of astrocyte stimulation in a pilocarpine-induced seizure model and a model of chronic neocortical epilepsy model with FCD, suggesting a general phenomenon. Currently, the theory of "excitability-inhibition" imbalance of signal transmission between neurons is the central dogma to explain the mechanism of epilepsy. However, intervention methods targeting "neuron-neuron" interactions, including anti-seizure drugs[46] and gene therapy approaches[47] often produce many side effects. Compared with general inhibition of neurons, optogenetic stimulation of astrocytes showed anti-seizure effects with several advantages: (1) wide therapeutic window, wherein astrocyte stimulation in seizure focus produce anti-seizure effect both in early and later phases; (2) large-space efficacy; as astrocyte stimulation even in remote cortical region can effectively attenuate the occurrence of GS, possibly via a gap junction related astrocyte network; and (3) with minimal effects on normal physiological functions, which may be due to astrocytic Na⁺-K⁺-ATPase-mediated activity-dependent inhibition of high-frequency firing pyramidal neurons in seizure (Supplementary Fig. 9).

Astrocytes modulate the excitability of neuronal networks through a multitude of pathways, among which Ca²⁺-dependent gliotransmitter release is the most classically studied[48–50]. Notably, several previous studies have indicated that the activation of astrocytes can potentiate seizure activity via Ca²⁺ mediated glial glutamate release[23–25]. In our study, there is indeed Ca²⁺ increase in astrocytes during seizure occurrence. Surprisingly, we found that Ca²⁺ increase is not involved in the anti-seizure effects of astrocyte stimulation, as two different types of Ca²⁺ blockers, the non-competitive inhibitor of the sarco/endoplasmic reticulum Ca²⁺ ATPase TG and the cell-permeant Ca²⁺ chelator BAPTA-AM, did not affect anti-seizure effects. Further, we found that chemogenetic activation in *GFAP-hM3Dq* mice selectively elevated intracellular Ca²⁺ in astrocytes and directly accelerated early seizure onset. This is consistent with a previous study which found that astrocytic Ca²⁺-mediated glutamate release can trigger or promote seizure activity[23–25]. Beyond Ca²⁺ signaling, we found that astrocytic Na⁺-K⁺-ATPase mediates the anti-seizure effects conferred by optogenetic stimulation of astrocytes. Application of the Na⁺-K⁺-ATPase inhibitor ouabain partially reversed the anti-seizure effects of astrocyte stimulation, whereas knockdown of astrocytic Na⁺-K⁺-ATPase completely reversed these effects during the whole process of seizure development. Knockdown of astrocytic Na⁺-K⁺-ATPase also reversed activity-dependent inhibition of high-frequency firing of pyramidal neurons. In seizure status characterized by excessive K⁺ accumulation in the extracellular microenvironment[51,52], astrocyte stimulation likely

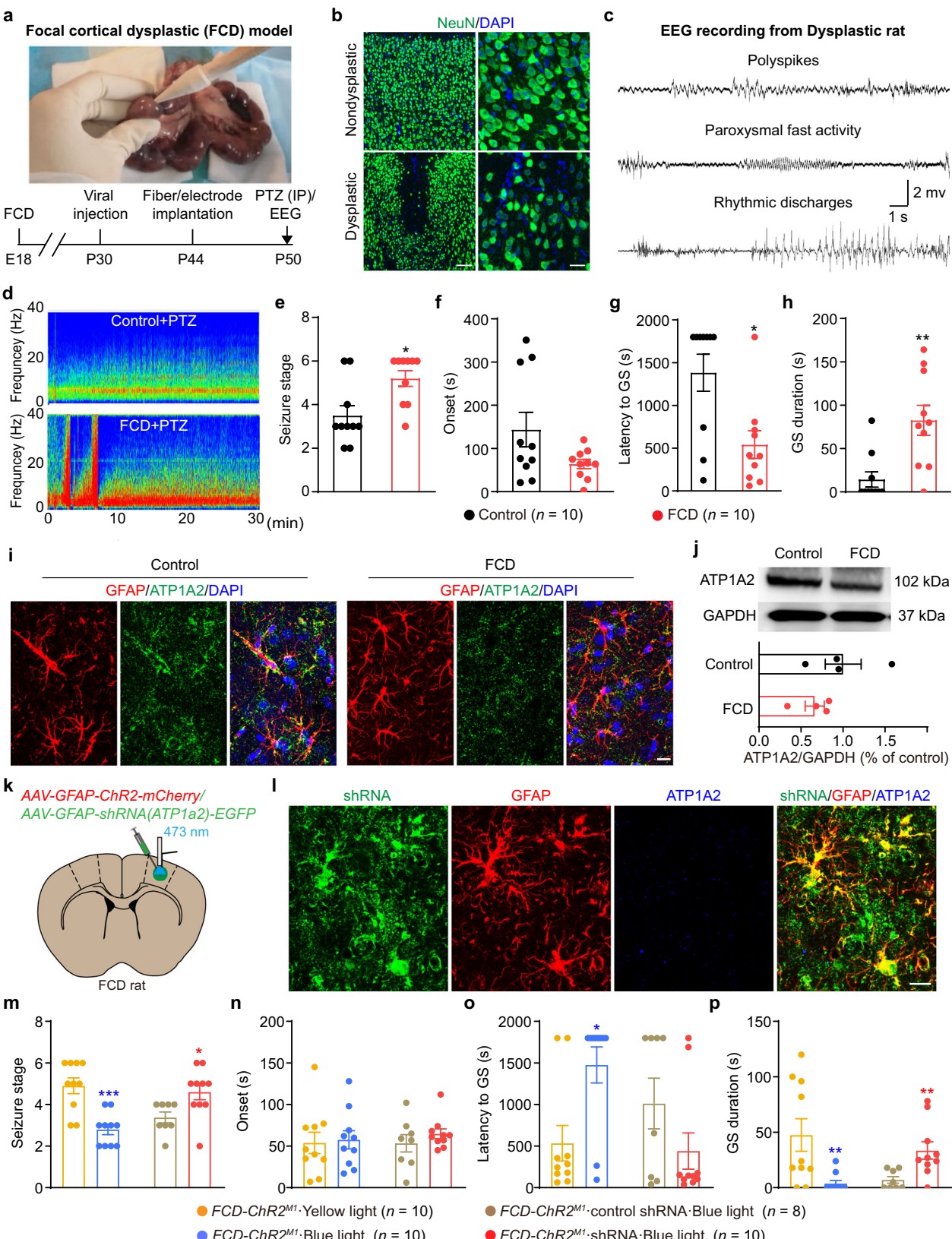

does not substantially increase intracellular $Ca^{2+}$, as intracellular $Ca^{2+}$ levels are already high. Instead, astrocyte stimulation increases intracellular $Na^+$, thereby activating the astrocytic $Na^+$-$K^+$-ATPase to permit uptake of extracellular $K^+$. Rescue of excessive extracellular $K^+$ reduces neuronal hyperexcitability of pyramidal neurons, ultimately attenuating seizure activity. Thus, our findings indicate that there may be a two-faced function of activated astrocytes in seizure conditions: one is $Ca^{2+}$ mediated astrocytic activation that is involved in seizure initiation in the early stage; the other is $Na^+$-$K^+$-ATPase-mediated astrocytic activation that may be more involved in the whole process of seizure development and act as an intrinsic anti-seizure mechanism. In fact, gene mutations in the $Na^+$-$K^+$-ATPase are critical in modulating the

**Fig. 6 | Optogenetic-driven astrocytic Na⁺-K⁺-ATPase rescues seizure susceptibility of FCD rats. a** Establishment of FCD model in SD rats and timeline design for behavioral experiments. **b** Representative immunostaining images showed NeuN⁺ cells in nondysplastic (control) and dysplastic (FCD) cortex. Scale bar, 50 μm and 25 μm. **c** Representative cortical EEG traces recorded from dysplastic FCD rat showed several forms of spontaneous epileptic waves. **d** Representative energy spectra from control and FCD rats injected with PTZ. **e**–**h** Seizure susceptibility of control and FCD rats was tested after PTZ injection (i.p. injection, 60 mg/kg) and seizure stage (**e**), EEG onset (**f**), latency to GS (**g**) and GS duration (**h**) were recorded. **i** Fluorescent images showed the expression pattern of Na⁺-K⁺-ATPase protein (ATP1A2) in control and FCD cortical tissue. **j** Representative western blot bands and quantification of Na⁺-K⁺-ATPase protein (ATP1A2) expression in control and FCD cortex. GAPDH was used as the internal control. **k** FCD rats were injected with ChR2-EYFP and/or pAAV-shortGFAP-MCS-EGFP-3FLAG-mir30shRNA(Atp1α2) in the M1 and implanted with fiber and electrode for blue light stimulation and EEG recording. **l** Fluorescent images showed shRNA(Atp1α2) co-localization with GFAP⁺ astrocytes and knockdown of astrocytic Na⁺-K⁺-ATPase protein (ATP1A2) expression in FCD cortical tissue. Scale bar, 20 μm. **m**–**p** Effects of optogenetic stimulation of astrocytes on seizure stage (**m**), EEG onset (**n**), latency to GS (**o**) and GS duration (**p**) in *FCD-ChR2^MI* rats, in the condition of astrocytic Na⁺-K⁺-ATPase inhibition by shRNA-knockdown. *$p < 0.05$, **$p < 0.01$, ***$p < 0.001$ compared with each control group. Data shown as mean ± s.e.m. The number of mice used is indicated in figures. For detailed statistical information, see Supplementary Data 1. Uncropped version of western blots that labeled with the relevant panel and antibody were provided in Source data western blot. Source data are provided as a Source Data file.

epilepsy phenotype[53–55], thus enhancing the astrocytic Na⁺-K⁺-ATPase is a promising future approach for epilepsy treatment. Indeed, we found that an optogenetic-driven astrocytic Na⁺-K⁺-ATPase rescues seizure susceptibility of FCD model, a chronic intractable neocortical epilepsy model. Additionally, both the Na⁺/Ca²⁺ exchanger and the Na⁺/H⁺ exchanger have been reported to be involved in seizure modulation[56,57]. Since both exchangers play an important role in Na⁺ homeostasis[20], they might also be driven to function in response to the Na⁺ enhancement caused by ChR2 activation. Whether they contribute to the anti-seizure effect of optogenetic activation of astrocytes needs further investigation.

In summary, our study expands the neuro-centric view of epilepsy pathogenesis to include multifaceted roles for activated astrocytes. Importantly, we uncover a promising anti-seizure strategy with optogenetic control of astrocytic Na⁺-K⁺-ATPase activity, providing alternative ideas and a potential target for the treatment of intractable epilepsy. Given recent studies which have greatly expanded our understanding of regional and phenotypic astrocyte heterogeneity[58,59], future studies exploring whether and how astrocytes may undergo phenotypic changes to contribute to seizure pathology or resolution are warranted. An expanded understanding of astrocyte biology and heterogeneity and the involvement of these cells in pathogenesis may lead to effective strategies to treat intractable epilepsy.

## Methods
### Animals
Adult (males, 2–4 months) *GFAP-Cre* mice, *CaMKIIα-Cre* mice and Sprague Dawley (SD) rats were used in this study. *GFAP-Cre* transgenic mice, which contain the 2.2-kB fragment of the human GFAP promoter driving expression of Cre recombinase, surrounded by four copies of genomic insulator, were generated by Dr. K.D. McCarthy at the University of North Carolina at Chapel Hill. Adult *GFAP-Cre* mice express Cre recombinase protein in astrocytes but not in neurons or oligodendrocytes[11,14]. *CaMKIIα-Cre* transgenic mice, which have the mouse calcium/calmodulin-dependent protein kinase II alpha (CamkIIa) promoter driving Cre recombinase expression in the forebrain excitatory neurons in the hippocampus and cortex[60], were from The Jackson Laboratory (Stock No: 005359). Two to five animals were housed in each cage on a 12-h light/dark cycle with food and water ad libitum, and they were individually housed after surgery. The ambient temperature in animal facility was kept about 23–26 °C and humidity was about 50–60%. No statistical method was used to predetermine sample size; sample sizes were estimated based on our previous studies for similar types of behavioral, biochemical, and electrophysiological analyses[61,62]. All behavior experiments were conducted between 9:00 and 17:00 in a blinded manner. All experimental procedures performed complied with the Zhejiang University Animal Experimentation Committee and were in complete compliance with the National Institutes of Health Guide for the Care and Use of Laboratory Animals.

### Stereotactic virus injections
Mice were anesthetized with pentobarbital sodium (50 mg/kg, i.p.) and were mounted on a stereotaxic apparatus (512600, Stoelting Co.). A small volume of virus solution (500 nl) was injected into the primary motor cortex, or M1, (Mouse: AP: +1.0 mm; ML: ±1.6 mm; V: −1.6 mm; Rat: AP: +1.56 mm; ML: ±2.5 mm; V: −2.0 mm) at a rate of 100 nl/min using a microsyringe pump (Micro4, World Precision Instruments). The injection needle was withdrawn 10 min after the infusion. The coordinates were measured from the Bregma, according to the atlas of Franklin and Paxinos[63]. Behavioral tests and in vivo electrophysiological recordings were conducted at least 3 weeks following virus injection. The accurate position of injection sites was confirmed in all animals post hoc by preparing sections (30 μm) containing motor cortex. Only the mice with correct locations of viral expression were taken into analysis.

For selective manipulation of astrocytes, *GFAP-Cre* mice were injected with AAV-EF1a-DIO-hChR2(H134R)-mCherry/EYFP virus or AAV-EF1a-DIO-hM3D(Gq)-mCherry virus. For astrocytes calcium signaling recording, *GFAP-Cre* mice were injected with AAV-Ef1α-DIO-GCaMP6m-GFP virus. For selective knockdown of astrocytic Na⁺-K⁺-ATPase, we designed the interference virus AAV-shortGFAP-MCS-EGFP-3FLAG-mir30shRNA(Atp1α2). The sequence used for the shRNA (Atp1α2) is: 5′-gca tca tat cag agg gta acc-3′. For selective inhibition of pyramidal neurons, *CaMKIIα-Cre* mice were injected with AAV-CAG-FLEX-ArchT-tdTomato virus. All control mice were injected with the corresponding control virus and had the same treatment. AAVs used in this study were purchased from OBiO and stored at −80 °C.

### Cannula and electrode implantation surgery
Animals were anesthetized with pentobarbital sodium (50 mg/kg, i.p.) and mounted on a stereotaxic apparatus (512600, Stoelting Co.). Optical fiber (200 μm, NA: 0.22/0.37, Newdoon Inc.) was stereotaxically implanted into primary motor cortex (M1) (Mouse: AP: +1.0 mm; ML: ±1.6 mm; V: −1.2 mm; Rat: AP: +1.56 mm; ML: ±2.5 mm; V: −2.0 mm) to enable light delivery. A custom-made cannula-electrode which was made of a cannula (62003, RWD Life Science Co., Ltd) tied to a stainless-steel electrode (791500, A-M Systems) was stereotaxically implanted into the primary motor cortex M1 (Mouse: AP: +1.0 mm; ML: ±1.6 mm; V: −1.6 mm; Rat: AP: +1.56 mm; ML: ±2.5 mm; V: −2.0 mm) to enable drug injection, light delivery and EEG recording. The cannula was kept ~0.5 mm above the electrode tips. Two screws were placed in the skull over the cerebellum to serve as the reference and ground electrode for EEG recording. The experiments were started after 1 week of recovery. When the behavioral experiments were completed, all experimental mice were required to confirm the implantation position of electrode and cannula. Only the correct one could be used for the result analysis.

### Light stimulation parameters
Blue/yellow laser light (473/590 nm) was delivered through a 200 μm diameter optic fiber with NA 0.22/0.37 (NA 0.22 for behavior test and

NA 0.37 for calcium signaling recording) connected to an intelligent Optogenetic System (Newdoon Inc.). The optic fiber was inserted to the cannula directly above the motor cortex after drug injection. The blue light parameters were 1, 5 or 20 Hz with 10 ms/pulse; the yellow light used was constant. The laser light power was 5 mW, and it was applied in 30 s on/off cycles during the behavioral experiments.

## Intra-cortical KA model

After surgery recovery, KA (300 ng in 600 nl sterile saline, ab120100, Abcam) was injected into M1 with an infusion needle (62203, RWD Life Science) through the guide cannula to a depth of 1.6 mm. After the injection was completed, the needle was retained for 3 min and then blue/yellow light was delivered. EEG was recorded for a few minutes in freely moving mice before KA injection using a PowerLab system with Bio Amplifiers (N12128, ADInstruments) and LabChart 7 software at a sampling rate of 1 kHz. Seizure behavior within 90 min was observed and recorded, including the seizure stage and duration. Seizure severity was scored as follows: 1, mouth and facial movement. 2, head nodding. 3, forelimb clonus. 4, rearing with forelimb clonus. 5, rearing and falling with forelimb clonus. 6, fully tonic-clonic seizure. Seizure stage 1–3 were focal seizures (FS), and seizure stage 4–6 were GS. If the mouse presented status epilepticus, it is also considered to have a seizure stage of 6. If there was no evidence of a behavioral seizure, the seizure was precluded from a GS number. For each mouse, the highest seizure stage within 10 min, the highest seizure stage within 90 min, the latency to first EEG seizures, the latency to GS and the number of GSs were recorded. If no GS occurred during 90 min, the latency to GS was recorded as 90 min.

To determine whether astrocytic $Ca^{2+}$, $Na^+$-$K^+$-ATPase, Kir4.1 and gap junctions are involved in the anti-seizure effect of optogenetic stimulation of astrocytes, 500 nl thapsigargin (TG, 1 µM in sterile normal saline, T9033, Sigma), 500 nl BAPTA-AM (200 µM in sterile normal saline, A1076, Sigma), 500 nl ouabain (OUA, 100 µM in sterile normal saline, ab120748, Abcam), 500 nl VU0134992 (100 µM in 5% DMSO/sterile normal saline, HY-122560A, MCE) and 500 nl carbenoxolone (CBX, 1 mM in sterile normal saline, C4790, Sigma) was injected into the right M1 before KA injection. For chemogenetic activation of astrocytes, 1 mg/kg clozapine N-oxide (CNO, ab141704, Abcam) was intraperitoneally injected 15 min before KA injection.

## Intra-cortical pilocarpine model

After surgery recovery, 200 nl pilocarpine (5 M in sterile normal saline, ab141301, Abcam) was inject to M1 with an infusion needle (62203, RWD Life Science Co., Ltd) through the guide cannula to a depth of 1.6 mm. After 3 min of injection, blue light was delivered. EEG were recorded for a few minutes before the pilocarpine injection. Seizure behaviors within 90 min were observed and recorded, including the seizure stage and duration. Seizure severity was scored as mentioned in the KA model. For each mouse, the latency to GS, death rate, the mean frequency of epileptic spikes and EEG power in each 10 min interval were analyzed by LabChart 7 software.

## Focal cortical dysplasia model

Pregnant SD rats were used for this experiment. Rats were anesthetized with pentobarbital sodium (50 mg/kg, i.p.) at 18 days post-conception (E18). Bilateral uteri were extracted from the abdominal cavity through a longitudinal incision and a metal probe with a hemispheric tip that had a diameter of 2.3 mm, cooled by liquid nitrogen, was placed onto the scalp of a rat embryo from outside of the uterus wall for 5 s. Each embryo received two points of freeze lesion, located to the left and right of the cranial suture. After the operation, uteri were returned into the abdominal cavity and the cavity was filled with 0.9% saline. The peritoneum, muscles and skin were sutured. Rats were returned to their cages after they recovered from the anesthesia. Rat pups were born at E22 and reared by the mother rats.

For selective manipulation of astrocytes, P30 FCD rats were injected with AAV-GFAP-hChR2-EYFP in primary motor cortex (M1) (AP: +1.56 mm; ML: ±2.5 mm; V: −2.0 mm). For selective knockdown of astrocytic $Na^+$-$K^+$-ATPase during astrocytes activation by ChR2, P30 FCD rats were injected with AAV-GFAP-hChR2-mCherry and AAV-shortGFAP-MCS-EGFP-3FLAG-mir30shRNA (Atp1α2).

After 2 weeks of viral injection, recording electrodes and fibers for EEG recording were placed on the M1 region, which was then stimulated with light (AP: +1.56 mm; ML: ±2.5 mm; V: −2.0 mm). After another 1-week surgery recovery, pentylenetetrazol (PTZ, 60 mg/kg, i.p.) was injected and blue light (473 nm, 20 Hz, 5 mW, 10 ms, 30 s on/off duty cycle) was delivered. EEG were recorded for a few minutes before the PTZ injection by the PowerLab system with Bio Amplifiers (N12128, ADInstruments) and LabChart 7 software at a sampling rate of 1 kHz. Seizure behaviors within 30 min were observed and recorded, including the seizure stage and EEG duration of seizure. Seizure severity was scored as follows: 1, mouth and facial movement. 2, head nodding. 3, forelimb clonus. 4, rearing with forelimb clonus. 5, rearing and falling with forelimb clonus. 6, fully tonic-clonic GS. Seizure stage 1–3 were FSs, seizure stage 4–6 were GSs. For each rat, the highest seizure stage within 30 min, the latency to first EEG seizures, the latency to GS and the GS duration were recorded. If no GS occurred during 30 min, the latency to GS was recorded as 30 min.

## EEG analysis

As we previously reported[64], seizure events were defined electrographically as a spike in frequency (≥2 Hz), high amplitude (>3X baseline) rhythmic epileptiform activity with a minimal duration of 10 s. Seizure events separated by >10 s were deemed to be separate. Six seizure stages were defined as above and divided into two categories (FS, GS) based on the behavior, the seizure score, and whether or not there was post seizure depression in the EEG. Bursts of high-voltage sharp waves with no obvious convulsive behaviors were interpreted as non-convulsive FS. Typical paroxysmal EEG activity with post seizure depression along with obvious convulsive behaviors were interpreted as GS. Spectral analysis using the fast Fourier transform and quantitative analysis of EEG energy content were performed by LabChart 7 software.

## Open field test

Locomotor function was evaluated by open field test in *GFAP-ChR2$^{M1}$* and *CAMKII-ARCH$^{M1}$* mice. Mice were exposed to a square open arena (45 cm × 45 cm) with an opaque base and opaque walls (45 cm high). Each mouse was allowed 15 min to explore the area and its activity was recorded and analyzed by ANYMZE 4.98 software (Stoelting Co.). Blue or yellow light was delivered exactly as in the epilepsy behavior test. The surface was cleaned with 70% ethanol after each mouse was tested.

## In vivo single-unit recording and analysis

To verify the response of pyramidal neurons during ChR2-expressing astrocyte stimulation with blue light, neural activity was recorded and analyzed in urethane (1.4 g/kg, i.p.) anesthetized *GFAP-ChR2$^{M1}$* mice by in vivo single-unit recording. Body temperature of the mouse was kept constant at 37 °C using a heating pad. Craniotomy was conducted in primary motor cortex (M1) (AP: +1.0 mm; ML: ±1.6 mm; V: −1.6 mm). A stainless-steel screw was screwed into the cerebellum to connect the electrode as a common reference. A bundle microelectrode of 8–12 nichrome wires coated by formvar (25-µm Diameter, 761500, A-M systems) with impedance of 1–2 M, was sticked with or without optical fiber for neuronal activities recording and light stimulation. The microelectrode and fiber were lowered into the brain until they were just above the target structure M1 and advanced slowly using hydraulic microdrives (Narishige International USA, Inc.) to isolate single cells. After the neuron had stable firing, we recorded the baseline for 1 min and then blue light (473 nm, 5 mW, 10 ms, 20 Hz, 10 s on/off cycle) was

delivered to manipulate ChR2-expressed astrocytes for the next 3 min. When the light stopped, we continued recording for another 3 min. To record ipsilateral neuronal activities in the KA-induced seizure condition, we injected KA into M1 before the recording.

The signals were sampled by the Cerebus 6.04 system (Blackrock Microsystems) with a sampling rate of 30 kHz, a high-pass filter at 250 Hz and a requirement that the signal to noise ratio be greater than 3:1. Offline sorting software (v4, Plexon Inc) was used to confirm the quality of the recorded cells. Peri-event raster, peri-event histogram, and autocorrelation analysis were used to sort neuronal data by Neuroexplorer 4.0. (Nex Technologies). Putative pyramidal neurons were identified by low firing rate (≤10 Hz), wide spike waveform (≥0.3 ms), and sharp autocorrelogram. As pyramidal cells occasionally fire bursts of spikes at short interspike intervals, there was a fast exponential decay with several msec from peak in the autocorrelograms. For analysis, firing rates across time were analyzed by averaging the spike times within 20 s intervals. The criteria used to define an "excited" or "inhibited" neuronal response were as follows: firing rates were considered to be significantly different if they were >2 SDs greater or less than baseline averages[65]. Neurons were categorized by their change in firing rate during light stimulation into the following three groups: (1) Activation, (2) Inhibition, or (3) No response. After experiments, all recorded animals were perfused for histological verification.

### Fiber photometry

KA was injected into M1 of *GFAP-GCaMP6m* mice through the cannula to induce seizure and then a fiber was inserted to capture the calcium signaling during the seizure using a fiber photometry system (Thinkertech Nanjing Bioscience Inc, Co., Ltd.). To verify the effect of $Ca^{2+}$ in this process, BAPTA-AM and TG were injected into M1 before KA injection. To record the calcium signaling during chemogenetic activation of astrocytes, we injected mixed AAV-Ef1α-DIO-GCaMP6m-GFP and AAV-Ef1α-DIO-hM3D(Gq)-mCherry virus to M1. We recorded the baseline calcium signaling for 10 min and then intraperitoneally injected 1 mg/kg CNO. After recording, photometry data were exported to MATLAB Mat files (R2017b, MathWorks, USA) for further analysis. We segmented the data based on KA or CNO injection and derived the values of fluorescence change ($\Delta F/F$) by calculating $(F - F0)/F0$[65].

### In vitro electrophysiology

*GFAP-ChR2* mice were anesthetized and then decapitated. The brain was quickly transferred to ice-cold sucrose-based ACSF containing (in mM): Sucrose 75, NaCl 87, KCl 2.5, $NaH_2PO_4$ 1.25, $CaCl_2$ 0.5, $MgCl_2$ 7, $NaHCO_3$ 26 and glucose 25. Cortical slices (300 μm) were cut using a vibratome (VT1000S, Leica). Subsequently, slices were transferred to normal ACSF solution (in mM): NaCl 124, KCl 3, $NaH_2PO_4$ 1.25, $CaCl_2$ 2, $MgSO_4$ 1, $NaHCO_3$ 26, glucose 10. After recovery for 30 min at 34 °C in ACSF, the slices were incubated at room temperature for 1 h. All extracellular solutions were constantly carbogenated (95% $O_2$, 5% $CO_2$).

Whole-cell recordings of ChR2-expressing astrocytes were performed at room temperature with an Axon MultiClamp 700B Amplifier (Molecular Devices). Data were low-pass-filtered at 3 KHz and sampled at 10 KHz. Patch pipettes were filled with a solution containing the following (in mM): KCl 140, $CaCl_2$ 0.5, $MgCl_2$ 1, EGTA 5, HEPE 10, Mg-ATP 3, and Na-GTP 0.3 (pH 7.2–7.3, adjusted with KOH; 288 mOsm). In voltage-clamp mode, whole-cell patch-clamped astrocytes were held at −80 mV. During the recording, baseline was recorded for 2 min and then blue light was delivered (20 Hz) for 2 min. After the blue light stimulation, we continued to record the post-line for another 2 min. RMP of these astrocytes was analyzed before and after blue light stimulation. Data acquisition and analysis were performed with Axon pClamp 10.3 software (Molecular Devices).

### Extracellular K⁺ measurement

We used microdialysis for sample collection[66,67]. Briefly, *GFAP-ChR2^M1* mice were anesthetized with pentobarbital sodium (50 mg/kg, i.p.) and mounted on a stereotaxic apparatus. Two guide cannulas (Carnegie Medicine) were stereotaxically implanted 1.5 mm apart in primary motor cortex M1, one for sample collection and the other one for light delivery. The screws and the guide cannulas were affixed together to the skull with cyanoacrylate and dental acrylic. After 7 days of recovery, microdialysis probes (CMA 7 Microdialysis Probes, 1 mm membrane length) were lowered through the guide cannula to a depth of 1.6 mm according to the bregma, and the mouse was placed in a freely moving system, consisting of a plastic cylinder with a counter-balancing arm carrying a 2-channel swivel (CMA Microdialysis). After balancing for 2 h after insertion, perfusion of the probe was started with potassium-free Ringer's solution with equimolar sodium chloride. The inflow to the probe was driven by a CMA/100 microinjection pump (flow rate was 0.5 μl/min), and outflow samples were collected continuously. The one baseline dialysate sample was collected over a period of 60 min before the mice were injected with KA. Following KA administration, samples were collected over the next 1 h with/without optogenetic stimulation of astrocytes. The in vitro recovery rate of the dialysis system for potassium was determined before the experiment in Ringer's solution containing potassium chloride (1–20 mM). The concentrations of K⁺ in each dialysate were detected by inductively coupled plasma atomic emission spectroscopy[31].

### Immunohistochemistry

After appropriate survival times, animals were deeply anesthetized with pentobarbital sodium (100 mg/kg, i.p.) and perfused with normal saline, followed by 4% paraformaldehyde (PFA). After the perfusion, the brain was removed and post-fixed in 4% PFA overnight and then dehydrated in 30% sucrose for 48 h. Brain tissue sections (30 μm) were cut in a cryostat (CM 3050S, Leica). After rinsing with 0.5% Triton-X 100 in PBS for 30 min, brain sections were blocked with 5% donkey serum for 2 h at room temperature and then incubated overnight at 4 °C with the following primary antibodies: anti-GFAP antibody (rabbit, 1:400, BA0056, Boster), anti-GFAP antibody (mouse, 1:400, G3893, Sigma), anti-NeuN (rabbit, 1:400, MABN140, Millipore), anti-ATP1A2 antibody (rabbit, 1:300, PA5-77512, Invitrogen). After washing, the sections were incubated with Alexa Fluor™ 488/594/647 secondary antibody (1:1000, Molecular Probes) for 2 h at room temperature. After rinsing the sections, we mounted the sections on slides using Vectashield Mounting Media (Vector Labs). The stained sections were examined with a Leica SP8 confocal microscope.

### Western blotting

Cortical homogenate (60 μg) was resolved with 12.5% SDS-PAGE and transferred to a nitrocellulose membrane, which was then blocked with 5% nonfat milk in PBS for 1 h. Then the membranes were incubated with the primary antibodies anti-ATP1A2 mAb (1:500, Abcam) and anti-GAPDH (1:5000; Kang-chen). After incubation with secondary antibodies goat anti-Rat/Mouse IgG (H + L) HRP (GRT007/GAM007, Multisciences) for 2 h, proteins were visualized by chemiluminescence with an ECL Western Blotting Substrate Kit (61809, Sungky BioTech) according to the manufacturer's instructions. The immunoreactivity of individual bands on western blots was measured by Image-Pro Plus 7 software.

### Statistical analyses

All the data were expressed as mean ± s.e.m. Number of experimental replicates (*n*) is indicated in the figure legends and refers to the number of experimental subjects independently treated in each experimental condition. Statistical analyses were completed with Prism GraphPad 8.0 using appropriate methods as specified in each figure's legend. All detailed statistic parameters are showed in

Supplementary Data 1. The criterion for statistical significance was two-tailed $p < 0.05$.

## Reporting summary

Further information on research design is available in the Nature Portfolio Reporting Summary linked to this article.

## Data availability

All data supporting the findings of this study are available within the paper and its Supplementary Information files "Source Data". Source data are provided with this paper.

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

## Acknowledgements

This project was supported by grants from the National Key R&D program of China (2020YFA0803902), the National Natural Science Foundation of China (82022071, 81630098, 81973298 and 81671282) and Natural Science Foundation of Zhejiang Province (LD22H310003). We are very grateful to Dr. Christopher R Donnelly for language editing.

## Author contributions

Y.W., Z.C. and J.Z. designed the project, interpreted the data and wrote the paper with input from all authors. J.Z., Y.W., J.S., Yang Z., F.F., Y.S. and Y.L. performed all experiments. C.X. and X.L. contributed to the in vitro electrophysiology. Yanrong Z. contributed to cell culture. S.W., Y.R., J.L. and S.D. contributed to interpretation of data. Z.C. and Y.W. supervised all aspects of the work.

## Competing interests

The authors declare no competing interests.
