## [Peer Review File · Nature Communications]

Activated astrocytes attenuate neocortical seizures in animal models through driving Na⁺-K⁺-ATPaseREVIEWER COMMENTS

Reviewer #1 (Remarks to the Author):

This is an impressive study when one looks at the totality of data. However, is it impactful or not? Is it largely confirming long known facts? As an astrocyte biologist I am struggling to come up with the definitive answer. Let me explain.

Involvement of glia in the etiology of certain forms of epilepsy have been suggested for over 50 years, starting with studies by Pollen and Trachtenberg (1970) in vivo and later human mesiotemporal lobe epilepsy. Many studies by Uwe Heinemann and colleagues (70s-80s) established that extracellular K^+ rises to 15 mM during epilepsy and Ransom and colleagues showed that K^+ is maintained by both astrocytes and importantly the neuronal ATPase (1995). Astrocytes at the scar have been shown to have decreased Kir4.1 channels and loss of gap junctions pointing towards extracellular K^+ and being involved in seizure genesis. Blocking K^+ channels with 4-AP is one way to induce seizures. Hence the notion that one could reduce seizures threshold if extracellular K^+ could be maintained at a physiological level and that this would be "anti-epileptic" will not surprise anyone in the epilepsy field.

In fact, if one lowers K^+ to lower than physiological level one can completely quit down the entire brain.

But how to do it? How to stimulate neuronal or glial K^+ uptake. It is well established that the neuronal ATPase responds to even small changes in intracellular Na^+ and takes up K^+ . The same is true, albeit to a slightly lesser extend for the glial ATPase. The authors of this study inserted a light activated cation channel into astrocytes. Flick on the light, Na^+ enters the astrocytes and the ATPase will start its work pumping K^+ in, hyperpolarizing neurons and astrocytes and quieting the brain. That is an experimental situation that bears no resemblance to actual physiology. However, the study confirms that reducing K^+ from the extracellular space is indeed reducing hyperexcitability. But it proofs what we had long know from studies that measured extracellular K^+ using microelectrodes. Yet this study found a way to experimentally coax the astrocytes into taking up K^+ a function normally performed only by the neuronal ATPase as only neurons see a physiological increase in Na^+ as they fire action potentials. Unfortunately, we do not have a way to selectively activate the astrocytic ATPase in humans with epilepsy in a similar way and likely will not express opsins in patients any time soon.

□ Most importantly, however, we have not learned any new information on the etiology of epilepsy. What causes neurons to be hyperexcitable?

□ Secondly, we have not gained a better understanding of what astrocyte have to do with epilepsy. We only learned that if we can make them pump more K^+ out of the extracellular space than they ordinarily would it quiets the brain and is beneficial in this scenario. Indeed, you maybe able to accomplish the same by expressing opsins in other brain cells (oligodendrocytes, microglia, endothelial cells)

The paper uses a series of compelling experiments to make the case that the optogenetically induced anti-seizure effect is due to ATPase function in astrocytes. If that's what the paper is all about, these findings are sound I only find a few editorial things wrong with it which I list below. When it comes to making any new mechanistic conclusions regarding astrocytes and epilepsy, I don't know what they are and how they would be substantiated by the paper.

So in many ways the newsworthiness to a general audience even a gliocentric one is limited, yet the magnitude of the data provided is quite stunning.

However, the authors repeatedly state that their findings provide novel mechanistic insight into the role for astrocytes in epilepsy which is simply not the case.

Editorial:

- The paper is dense and very difficult to read.
- Various transgenic animals are used and not properly introduced.
- Figures are not at all described in the text and some refer in the text to the wrong figure.
- There is insufficient raw data on electrical excitability. There are many summary graphs that make it difficult to ascertain the quality of the data.

Reviewer #2 (Remarks to the Author):

Summary:

In this manuscript, Zhao et al. reports astrocytes decrease hyperexcitability of pyramidal neurons under seizure conditions by maintaining optimal extracellular K level through NaKATPase. Overall, authors are investigating important topic of the role of astrocytes in regulating neuronal excitability using chemoconvulsant-induced seizure models. Although authors have utilized multiple approaches to test their hypothesis from various angles, there remains areas where additional data could strengthen the manuscript and better support the conclusions drawn. Methods section should also provide detailed information about various techniques used. Please refer to the following comments for more details.

Major issues:

1) It is mentioned on page 10 that ChR2 is a light-gated cation-selective membrane channel that mainly permits entry of Na⁺ and Ca²⁺ ions into cells. Please provide citation(s). My understanding is that ChR2 is also permeable to K⁺ and H⁺ as well. If K⁺ is as permeable to ChR2 as Na⁺, how would that affect the function of NaKATPase in astrocytes?

2) Fig 5 and Extended Data Fig 5: It is essential to include in vivo data on the level of ATP1A2 in astrocytes and neurons from the cortical tissue following injection of shRNA against ATP1A2. ATP1A2 expression level in neurons should also be provided since it is critical to show no effect on neuronal ATP1A2 following shRNA injection. Although authors include some IHC data from primary cortical cell culture pretreated with shRNA against APT1A2 to support knockdown of ATP1A2 in astrocytes, this data do not provide any meaningful information given the absence of astrocytic and neuronal markers as well as of the control group.

3) Describe in detail the process for collecting extracellular fluid from brain around KA injection site for the measurement of extracellular K⁺. Information provided in the manuscript is too brief. Was it through microdialysis?

4) Kir4.1 is primarily responsible for taking up excessive extracellular K under hyperexcitable conditions. Did you check the expression level of Kir4.1 in astrocytes? Could it have contributed to removal of extracellular K, instead or in addition to NaKATPase, following chemoconvulsant application?

5) One of the striking findings in this manuscript is a reduction in KA-induced seizures in response to optogenetic stimulation of ChR2-expressing astrocytes in the contralateral side (Fig 1l-q). This suggests the role of astrocytes in ionic buffering via gap junctions through astrocytic syncytium. Authors acknowledge this point and attempts to block GJ-mediated astrocytic coupling using carbenoxolone (CBX). However, it is important to include data on the effects of CBX on KA-induced seizures while stimulating astrocytes in contralateral hemisphere. Extended Data Fig 6 provides data acquired by applying all injections ipsilaterally. Other endpoints to check should be the level of extracellular K and the excitability of pyramidal neurons after injecting CBX and stimulating astrocytes in the contralateral side.

6) Fig 6: More information should be provided on seizure onset, frequency and severity in the FCD rats. Although Fig 6c shows examples of abnormal electrographic activity from dysplastic rats, this is not sufficient. Also, a single subthreshold dose of PTZ typically does not cause seizure, at least not a generalized seizure. I am not sure how rats in the FCD group would respond to a subthreshold dose of PTZ, but it should not cause seizure in control rats. However, panels 6d-g show PTZ-induced seizure occurrence in all control rats. Please clarify.

7) Since the FCD rats show spontaneous seizures (Fig 6c), it is desirable to include the data on the effects of optogenetic stimulation of ChR2-expressing astrocytes on seizures in this model (without using PTZ protocol).

8) Fig 6h: What was the level of NaKATPase in astrocytes? Also, membrane fraction of NaKATPase should be more relevant.

9) The level of NaKATPase should be measured in astrocytes from the cortical tissue injected with shRNA against ATP1A2 to confirm the knockdown of NaKATPase.

10) Authors rationalizes that ChR2 activation in astrocytes causes increase in Na intracellularly and that activates NaKATPase to exchange Na for K intracellularly, which in turn, reduces extracellular K concentration. That is certainly possible, but astrocytes also express other exchangers such as Na/Ca exchanger and Na/H exchanger which could play a role in regulating intracellular Na concentration in response to ChR2 activation. Authors should at least discuss these alternative possibilities.

11) Although Fig 1i-j shows no generalized seizures (GS) in GFAP-ChR2M1 mice (based on analyzing seizure data up to 90 min post-KA), most of these mice do develop GS eventually (about 150 min latency post-KA) as per Extended Data Fig 3. Does this suggest that the effects of optogenetic stimulation of ChR2-expressing astrocytes on KA-induced seizures are transient? Did you obtain EEG data from these mice for a longer term to compare spontaneous seizures?

Minor issues:

1) Fig 2I: Number of neurons in >5 Hz category is too low (n=2). To make any meaningful conclusion, it is desirable to have similar number of neurons (at least 5) recorded under all three firing rate categories. Also, the corresponding text in results section report 4 neurons in >5 Hz category. Please correct it.

2) Extended Data Fig 2e: Which frequency band is compared here?

3) Fig 5h-i: What does explain a huge difference in firing frequency of pyramidal neurons between KA+OUA and KA+shRNA conditions? OUA should inhibit NaKATPase expressed by both astrocytes and neurons, and that makes it inappropriate to test the role of astrocytic NaKATPase. However, its effects are quite comparable to shRNA-mediated knockdown of ATP1A2 in astrocytes.

4) Page 19: "Cannula and electrode implantation surgery" Line 8, Mention Rat before second set of coordinates

5) Page 21: Mention pilocarpine concentration used appropriately.

6) Page 12, line 7 (from bottom): in-line citation should be Fig 5I.

7) Methods section should include more details regarding EEG, in vivo single-unit recording, and slice electrophysiology.

Reviewer #3 (Remarks to the Author):

This manuscript reports that selectively activating Channel-rhodopsin (ChR2) in cortical astrocytes attenuates neocortical seizures. The authors show that this attenuation is independent of rapid Ca⁺⁺ signaling, which is confirmed to promote or facilitate seizure genesis consistent previous reports. Instead, the seizure attenuating effect of ChR2-activation of astrocytes is shown to be related to Na⁺-K⁺-ATPase-mediate K⁺ buffering. The findings both confirm and expand the growing body of information that networks of astrocytes connected via gap junctions play essential roles in buffering extracellular K⁺ and that astrocytes can thereby broadly and powerfully influence

neuronal excitability.

From a technical perspective I found the study well designed and well executed, with appropriate controls and sufficient replicates for statistical evaluations. The data are clearly presented, easy to follow, and are convincing. The data do not appear over-interpreted. The findings should be of broad interest regarding astrocyte-neuron interactions in general and adds specific mechanistic information to the long-standing notion that astrocytes are somehow involved in seizures. In particular, the findings point towards the potential for interventions that target astrocyte K⁺ buffering to potentially have relevance in the control of intractable seizures. For these reasons I think the study warrants exposure to a broad audience.

Specific comments

The specificity of the ChR2 etc targeting to astrocytes depends on the specificity of the GFAP-Cre mouse line used. The authors should provide more information in the methods or supplementary data about the mouse line used and the specificity of the targeting to astrocytes using this particular line. Although the authors show reporter targeting data in figure 1, they do not describe the source of the mouse line that they used other than to cite a paper in which they have used the mouse line previously. Going to that cited paper, there is again no specific description of the mouse line other than to refer to a paper from 2006, which states in its title that the mouse line targets GFAP to neuronal progenitors during brain development. At first glance, the use of this mouse seems odd and confusing. The authors should explain in the methods or supplementary data that in adult mice from this line, which is what they used to inject AAV, GFAP-Cre is expressed in the neocortex essentially only in astrocytes and that as their cell counts show, >98% of neurons did not express reporter.

Comments raised by peer reviewers in response to manuscript submission and author responses:

Reviewer #1 (Remarks to the Author):

Comment 1: *This is an impressive study when one looks at the totality of data. However, is it impactful or not? Is it largely confirming long known facts? As an astrocyte biologist I am struggling to come up with the definitive answer. Let me explain.*

Involvement of glia in the etiology of certain forms of epilepsy have been suggested for over 50 years, starting with studies by Pollen and Trachtenberg (1970) in vivo and later human mesiotemporal lobe epilepsy. Many studies by Uwe Heinemann and colleagues (70s-80s) established that extracellular K^+ rises to 15 mM during epilepsy and Ransom and colleagues showed that K^+ is maintained by both astrocytes and importantly the neuronal ATPase (1995). Astrocytes at the scar have been shown to have decreased Kir4.1 channels and loss of gap junctions pointing towards extracellular K^+ and being involved in seizure genesis. Blocking K^+ channels with 4-AP is one way to induce seizures. Hence the notion that one could reduce seizures threshold if extracellular K^+ could be maintained at a physiological level and that this would be “anti-epileptic” will not surprise anyone in the epilepsy field. In fact, if one lowers K^+ to lower than physiological level one can completely quit down the entire brain.

But how to do it? How to stimulate neuronal or glial K^+ uptake. It is well established that the neuronal ATPase responds to even small changes in intracellular Na^+ and takes up K^+ . The same is true, albeit to a slightly lesser extend for the glial ATPase. The authors of this study inserted a light activated cation channel into astrocytes. Flick on the light, Na^+ enters the astrocytes and the ATPase will start its work pumping K^+ in, hyperpolarizing neurons and astrocytes and quieting the brain. That is an experimental situation that bears no resemblance to actual physiology. However, the study confirms that reducing K^+ from the extracellular space is indeed reducing hyperexcitability. But it proofs what we had long know from studies that measured extracellular K^+ using microelectrodes. Yet this study found a way to experimentally coax the astrocytes into taking up K^+ a function normally performed only by the neuronal ATPase as only neurons see a physiological increase in Na^+ as they fire action potentials. Unfortunately, we do not have a way to selectively activate the astrocytic ATPase in humans with epilepsy in a similar way and likely will not express opsins in patients any time soon. Most importantly, however, we have not learned any new information on the etiology of epilepsy. What causes neurons to be hyperexcitable?

Secondly, we have not gained a better understanding of what astrocyte have to do with epilepsy. We only learned that if we can make them pump more K^+ out of the extracellular space than they ordinarily would it quiets the brain and is beneficial in this scenario. Indeed, you maybe able to accomplish the same by expressing opsins in other brain cells (oligodendrocytes, microglia, endothelial cells).

The paper uses a series of compelling experiments to make the case that the optogenetically induced anti-seizure effect is due to ATPase function in astrocytes. If that's what the paper is all about, these findings are sound. I only find a few editorial things wrong with it which I list below.

When it comes to making any new mechanistic conclusions regarding astrocytes and epilepsy, I don't know what they are and how they would be substantiated by the paper. So in many ways the newsworthiness to a general audience even a gliocentric one is limited, yet the magnitude of the data provided is quite stunning.

However, the authors repeatedly state that their findings provide novel mechanistic insight into the role for astrocytes in epilepsy which is simply not the case.

Response 1: We would like to express our sincere appreciation for your time to review our manuscript and the constructive comments on this study, which are all valuable and very helpful for improving our paper. We hope that the following responses according to your comments could address your concerns.

Firstly, we are sorry for the confusing and misleading descriptions. We totally agree with the reviewer that extracellular K^+ homeostasis has long been considered to be involved in seizure genesis. As the reviewer mentioned, the notion about extracellular K^+ homeostasis maintained by neuronal and astrocytic $Na^+-K^+-ATPase$ to control the seizure genesis is not surprised to people in the epilepsy field. We have altered the overstated description about “novel mechanism in astrocyte-neuron signaling underlying seizure” in our revised manuscript.

For the novelty of our study, we try to address another critical question from a therapeutic view: how to maintain the extracellular K^+ concentration and control the seizure in a safe and powerful way? We believe that our study has two novel findings listed in the following two aspects: 1) Optogenetic activation of channelrhodopsin-2 (ChR2)-expressing astrocytes can effectively attenuate the severity of seizures and activity-dependently inhibit high-firing pyramidal neurons during seizure. Meanwhile, optogenetic activation of astrocytes has the advantages of wide therapeutic window (whole phase stimulation, early phase and late phase stimulation), large spatial effect (remote cortical region stimulation) and minimal side effect (no locomotion deficiency), compared to the directly optogenetic activation of neurons. These findings indicated a promising anti-seizure treatment for epilepsy. 2) Anti-seizure effects of optogenetic activation of astrocytes do not rely on classical calcium signaling pathways in astrocytes, but instead are dependent on astrocytic $Na^+-K^+-ATPase$ function. These results are surprising because what we previously think about was that the astrocytic activation should increase the Ca^{2+} signaling which could potentiate seizure activity via Ca^{2+} mediated glial glutamate release mechanism (PMID: 16116433, PMID: 20405049, PMID: 24101472). This finding would also give an important hint: K^+ homeostasis should be taken into consideration for the modulation of astrocyte by using optogenetics, as many previous studies focus on Ca^{2+} signaling related function. Our findings point towards the potential for optogenetic interventions that target astrocyte K^+ buffering to potentially have relevance in the control of intractable seizures.

For the translational concern, we agree with you that we do not have a way to selectively activate the astrocytic $Na^+-K^+-ATPase$ by optogenetics in humans with epilepsy now. The ultimate goal for our basic research is discovering the mechanisms and approaches for translational treatment of human patients. It will be a very promising direction if people could develop the noninvasive methods to selectively activate astrocytic $Na^+-K^+-ATPase$ combined with automatically detection of extracellular K^+ concentration and high epileptic spiking and further used in human patients with epilepsy. Meanwhile, regarding to the optogenetic technology applied in other cells (oligodendrocytes, microglia, and endothelial cells). It is possible that the ChR2 expressing-oligodendrocytes/microglia/endothelial cells could accomplish K^+ buffering, which really deserves further investigation. For this important point, we have added more discussion in the revised manuscript. But our study at least indicated that strengthen the astrocytic $Na^+-K^+-ATPase$ under seizure condition might be a promising approach for epilepsy controlling.

Taken together, we really appreciate your valuable comments, and we can then think deeply about the future direction for our research. To make it more clearly, we have revised the overstated description about “novel mechanism” in our revised manuscript, and pay much attention to the therapeutic meaning in our study. Further, based on your comment, we also added more discussion for the future studies. All the revisions are highlighted by red color.

Editorial:

Comment 2: *The paper is dense and very difficult to read.*

Response 2: Thanks for this suggestion. Accordingly, we have edited the manuscript to make it more logical and easier to read. We also asked Dr. Christopher R Donnelly for help to check the grammar.

Comment 3: *Various transgenic animals are used and not properly introduced.*

Response 3: Thanks for this comment. We added more details about transgenic animals in the “**Animals**” section.

- *GFAP-Cre* transgenic mice were generated by Dr. K.D. McCarthy at the University of North Carolina at Chapel Hill, which contain the 2.2-kB fragment of the human GFAP promoter driving expression of Cre recombinase, surrounded by four copies of genomic insulator. Adult *GFAP-Cre* mice express Cre recombinase protein in astrocytes but not in neurons or oligodendrocytes (PMID: 16458536, PMID: 28128211).
- *CaMKII α -Cre* transgenic mice was purchased from Jackson Laboratory (Stock No: 005359), which have the mouse calcium/calmodulin-dependent protein kinase II alpha (CamkIIa) promoter driving Cre recombinase expression in the forebrain excitatory neurons in the hippocampus and cortex (PMID: 8980237).

Comment 4: *Figures are not at all described in the text and some refer in the text to the wrong figure.*

Response 4: Thank you for your careful review. We checked the text and every figure carefully and corrected as suggested in the revised manuscript.

Comment 5: *There is insufficient raw data on electrical excitability. There are many summary graphs that make it difficult to ascertain the quality of the data.*

Response 5: Thanks for this suggestion. We have added raw data on electrical excitability in **Fig. 2b**, which showed the online recording process (the signal to noise ratio is greater than 3:1) and offline sorting process in *in vivo* single unit recording. Meanwhile, to make it more clearly, we have added more detailed description in the revised method section.

Revised Fig. 2a-b

Reviewer #2 (Remarks to the Author):

Comment 1: *Summary: In this manuscript, Zhao et al. reports astrocytes decrease hyperexcitability of pyramidal neurons under seizure conditions by maintaining optimal extracellular K level through Na⁺-K⁺-ATPase. Overall, authors are investigating important topic of the role of astrocytes in regulating neuronal excitability using chemoconvulsant-induced seizure models. Although authors have utilized multiple approaches to test their hypothesis from various angles, there remains areas where additional data could strengthen the manuscript and better support the conclusions drawn. Methods section should also provide detailed information about various techniques used. Please refer to the following comments for more details.*

Response 1: We would like to express our sincere appreciation for your time to review our manuscript. Your valuable and constructive comments will surely reinforce the conclusions and rigor of our manuscript. Below we address each specific comment raised in a point-by-point manner. We hope that the following responses according to your comments could address your concerns.

Major issues:

Comment 2: *It is mentioned on page 10 that ChR2 is a light-gated cation-selective membrane channel that mainly permits entry of Na⁺ and Ca²⁺ ions into cells. Please provide citation(s). My understanding is that ChR2 is also permeable to K⁺ and H⁺ as well. If K⁺ is as permeable to ChR2 as Na⁺, how would that affect the function of Na⁺-K⁺-ATPase in astrocytes?*

Response 2: Thanks very much for this important question.

We are sorry for the confusing description. Generally, ChR2 (channelrhodopsin-2) is capable of conducting nonselective cation, including Na⁺, K⁺, Ca²⁺, and H⁺, passively flow across the cellular membrane upon illumination with blue light. ChR2 conducts cations across the membrane in both directions but along the electrochemical gradient of the transported ions (PMID: 22196724; PMID: 14615590). Although, H⁺ preference over any other ion. However, due to the low availability of H⁺, other cations may contribute significantly to the photocurrent. In homeostasis, cytoplasmic [Ca²⁺] under resting conditions is ~10⁻⁴ mM, 10⁴ times lower than [Ca²⁺] in the extracellular milieu (~1 mM). Intracellular [Na⁺] was ~15 mM, 10 times lower than [Na⁺] in the extracellular milieu (~150 mM). Thus, ChR2 mainly permits entry of Na⁺ and Ca²⁺ ions into cells and the K_m for Ca²⁺ is 18.3 mM in ChR2 whereas the K_m values for Na⁺ are higher than 100 mM (PMID: 19192197).

On the contrary, intracellular [K⁺] was ~150 mM, 30 times higher than [K⁺] in the extracellular milieu (~5 mM). We agree with the reviewer that K⁺ may also be permeable to ChR2 (from intracellular to extracellular). Oceau *et al.* reported that ChR2 (H134R) activation in astrocytes and neurons caused a transit increase in extracellular K⁺ to ~7.4 mM and ~8.8 mM respectively in physical condition (PMID: 31116972). While, in seizure status, the extracellular [K⁺] rises to a high level (up to 10-12 mM) (PMID: 832122), which may limit K⁺ to be released into extracellular space during ChR2 activation. Our data directly showed optogenetic activation of astrocytes largely decrease K⁺ in seizure state.

Taken together, in seizure state, although Na⁺/Ca²⁺ may both enter into intracellular space, the Na⁺ entry triggers Na⁺-K⁺-ATPase activation in astrocytes and further buffers extracellular K⁺, which is the main anti-seizure mechanism. Based on this comment, we have added new citation and some more detailed discussion in the revised manuscript.

Comment 3: Fig 5 and Extended Data Fig 5: It is essential to include *in vivo* data on the level of ATP1A2 in astrocytes and neurons from the cortical tissue following injection of shRNA against ATP1A2. ATP1A2 expression level in neurons should also be provided since it is critical to show no effect on neuronal ATP1A2 following shRNA injection. Although authors include some IHC data from primary cortical cell culture pretreated with shRNA against APT1A2 to support knockdown of ATP1A2 in astrocytes, this data do not provide any meaningful information given the absence of astrocytic and neuronal markers as well as of the control group.

Response 3: Thanks for this important question. Accordingly, we did the immunocytochemistry to test the level of ATP1A2 protein expression both in astrocytes and neurons from the cortical tissue following injection of shRNA against ATP1A2. As the Fig. 5b showed, the expression of ATP1A2 protein in GFAP⁺/shRNA(Atp1α2)⁺ astrocytes was largely reduced than that in GFAP⁺/shRNA(NC)⁺ astrocytes. Meanwhile, shRNA(Atp1α2)⁺ was seldom co-localized with NeuN⁺ neurons (Extended Data Fig. 5b), which is consistent with previous finding that α2 isoform is expressed primarily in neurons during embryonic development while it is primarily expressed in glial cells in the adult (PMID: 11950769). All these data demonstrated the astrocytic knockdown of Na⁺-K⁺-ATPase. Thus, we have included this new data in the revised manuscript.

Revised Fig. 5b

Extended Data Fig. 5b

Comment 4: Describe in detail the process for collecting extracellular fluid from brain around KA injection site for the measurement of extracellular K⁺. Information provided in the manuscript is too brief. Was it through microdialysis?

Response 4: We thank the reviewer for this comment. We added a “Extracellular K⁺ measurement” in “Methods section.”

Extracellular K⁺ measurement. We used the microdialysis for sample collection as previous studies (PMID: 1977896, PMID: 27137202). Briefly, GFAP-ChR2^{M1} mice were anesthetized with pentobarbital sodium (50 mg/kg, i.p.) and were mounted on a stereotaxic apparatus. Two guide cannulas (Carnegie Medicine) were

stereotaxically implanted 1.5 mm apart in primary motor cortex M1, one for sample collection and the other one for light delivery. The screws and the guide cannulas were affixed together to the skull with cyanoacrylate and dental acrylic. After 7 days of recovery, microdialysis probes (CMA 7 Microdialysis Probes, 1 mm membrane length) were lowered through the guide cannula to a depth of 2.0 mm according to the bregma, and the mouse was placed in a freely moving system, consisting of a plastic cylinder with a counter-balancing arm carrying a 2-channel swivel (CMA Microdialysis). After balancing for 2 h after insertion, perfusion of the probe was started with potassium-free Ringer's solution with equimolar sodium chloride. The inflow to the probe was driven by a CMA/100 microinjection pump (flow rate was 0.5 μ l/min), and outflow samples were collected continuously. The one baseline dialysate sample was collected over a period of 60 min before the mice were applied with KA injection. Following KA administration, samples were collected over the next 1 h with/without optogenetic stimulation of astrocytes. The *in vitro* recovery rate of the dialysis system for potassium was determined before experiment in Ringer's solution containing potassium chloride (1-20 mM). The concentrations of K^+ in each dialysate were detected by inductively coupled plasma atomic emission spectroscopy as our previous study (PMID: 32042163).

Comment 5: *Kir4.1 is primarily responsible for taking up excessive extracellular K^+ under hyperexcitable conditions. Did you check the expression level of Kir4.1 in astrocytes? Could it have contributed to removal of extracellular K^+ , instead or in addition to NaKATPase, following chemoconvulsant application?*

Response 5: Thanks very much for this important question. We agree with reviewer that Kir4.1 plays an important role in K^+ buffering under the hyperexcitable conditions. To test whether Kir4.1-mediated K^+ buffering contributed to the anti-seizure effects of optogenetic activation of astrocytes, firstly we checked the expression level of Kir4.1 in astrocytes. Similar as previous studies (PMID: 23603404, PMID: 23922547), we found that Kir4.1 was expressed both in resting and activated astrocytes, and Kir4.1 was primarily stained in astrocytes which typically show a stellate-shape and were specifically co-stained with astrocytes maker GFAP.

Then, we applied the Kir4.1 blocker (VU0134992, PMID: 29895592) in KA-induced seizures with/without optogenetic stimulation of ChR2-expressing astrocytes. As shown in **Extended Data Fig.6**, Kir4.1 blocker alone slightly aggravates the severity of seizures, as shown increased GS number than vehicle treated mice. This is consistent with previous study that genetic deletion of Kir 4.1 would inhibit uptake of K^+ and elevated seizure susceptibility (PMID: 17091490), suggesting Kir4.1-mediated K^+ buffering is involved in seizure modulation.

While Kir4.1 blocker could not reverse the anti-seizure effects of optogenetic stimulation of ChR2-expressing astrocytes. As we shown in Fig.2, optogenetic stimulation of astrocytes will activity-dependently inhibit the pyramidal neurons, and the inhibition effect was most dramatic in neuron with high frequency firing (associated with high K^+ level). Kir4.1 blocking will increase extracellular K^+ concentration which could still be buffered by Na^+ - K^+ -ATPase. This result suggests that Kir4 alone is involved in seizure genesis, but it is not required for the anti-seizure effects of optogenetic activation of astrocytes.

Thus, we have included this new data in the revised manuscript.

Extended Data Fig. 6

Comment 6: One of the striking findings in this manuscript is a reduction in KA-induced seizures in response to optogenetic stimulation of ChR2-expressing astrocytes in the contralateral side (Fig 11-q). This suggests the role of astrocytes in ionic buffering via gap junctions through astrocytic syncytium. Authors acknowledge this point and attempts to block GJ-mediated astrocytic coupling using carbenoxolone (CBX). However, it is important to include data on the effects of CBX on KA-induced seizures while stimulating astrocytes in contralateral hemisphere. Extended Data Fig 6 provides data acquired by applying all injections ipsilaterally. Other endpoints to check should be the level of extracellular K and the excitability of pyramidal neurons after injecting CBX and stimulating astrocytes in the contralateral side.

Response 6: Thank you very much for providing the suggestion to strengthen our findings. Large-space efficacy is indeed one of the striking findings in our manuscript. We have addressed this point in two revised experiments and added description in the revised manuscript:

(1) We tested the effect of CBX on KA-induced seizure while photo-stimulating astrocytes in contralateral hemisphere. As shown in **Extended Data Fig. 7f-j**, astrocytic gap-junction inhibitor CBX could also reverse the anti-seizure effect of optogenetic stimulation of astrocytes in the contralateral side.

(2) In addition, we also performed *in vivo* single unit recording to test the function of astrocytic gap-function in the modulation of neuronal firing during optogenetic stimulation of ChR2-expressing astrocytes in KA-induced seizures. As shown in **Extended Data Fig. 7k and l**, both ipsilateral and contralateral injection of CBX reverse the suppression of cortical pyramidal neurons firing induced by optogenetic stimulation of astrocytes, suggesting astrocytic gap-junction might contribute to the buffering of K⁺ and neural excitability in seizure condition.

Thus, based on your comments, we have included these new data in the revised manuscript as following:

Extended Data Fig. 7

Comment 7: Fig 6: More information should be provided on seizure onset, frequency and severity in the FCD rats. Although Fig 6c shows examples of abnormal electrographic activity from dysplastic rats, this is not sufficient. Also, a single subthreshold dose of PTZ typically does not cause seizure, at least not a generalized seizure. I am not sure how rats in the FCD group would respond to a subthreshold dose of PTZ, but it should not cause seizure in control rats. However, panels 6d-g show PTZ-induced seizure occurrence in all control rats. Please clarify.

Response 7: Thank you very much for this important point.

We did record the EEG of FCD rats 8 h/day for 7 days on P44-P50. We analyzed the EEG and found that FCD rats showed spontaneous epileptiform activity, but the frequency is very low (~4 times/8h). This at least indicated that FCD rat may increase neural excitability and capable of generating abnormal electrographic activity. Thus, we included the more description of abnormal electrographic activity for FCD rats.

To further reliably test the seizure susceptibility in FCD rat, we challenged the FCD rat by a low dose of PTZ (60 mg/kg). Generally, a dose of 80–100 mg/kg i.p. is used to induce tonic-clonic generalized seizures (GS) in normal rats (Models of Seizures and Status Epilepticus Early in Life, Stéphane Auvin, Astrid Nehlig, in Models of Seizures and Epilepsy (Second Edition), 2017). In this study, administration of 60 mg/kg PTZ induces GS in 30% (3/10) control rats, but induce GS seizures in 90% (9/10) FCD rats. We think this is an appropriate dose for seizure susceptibility testing. We agree with the reviewer, using “subthreshold” word is not suitable. To clarify

this point, we have changed the description in our revised manuscript. In addition, we added the EEG spectra from control and FCD rats with PTZ injection (**Fig. 6d**).

Comment 8: Since the FCD rats show spontaneous seizures (Fig 6c), it is desirable to include the data on the effects of optogenetic stimulation of ChR2-expressing astrocytes on seizures in this model (without using PTZ protocol).

Response 8: Thanks very much for this comment. As we described in response 7, we did record the EEG of FCD rats 8 h/day for 7 days on P44-P50. We analyzed the EEG and found that FCD rats showed abnormal electrographic activity with extremely low frequency (~4 times/8h). To test the effects of optogenetic stimulation of ChR2-expressing astrocytes on spontaneous seizures, we need to long-termly (for several hours) stimulating astrocytes with light, which might heat the tissue and thus induce cellular damage (PMID: 31209377). Thus, to further reliably test the seizure susceptibility in FCD rat, we challenged the FCD rat by a low dose of PTZ (60 mg/kg). To make it more clear, we have included more description of abnormal electrographic activity for FCD rats.

Comment 9: Fig 6h: What was the level of NaKATPase in astrocytes? Also, membrane fraction of NaKATPase should be more relevant.

Response 9: Thank you very much for this comment. Accordingly, we tested the expression of $\alpha 2$ isoform of $\text{Na}^+\text{-K}^+\text{-ATPase}$ protein by immunostaining with enlarged view in control and FCD cortical tissues. As shown in the revised **Fig. 6i**, the membrane fraction of $\alpha 2$ level was slightly decreased in FCD rats, compared to control rats. Thus, we have added this new data in the revised manuscript as following:

Revised Fig. 6i

Comment 10: The level of NaKATPase should be measured in astrocytes from the cortical tissue injected with shRNA against ATP1A2 to confirm the knockdown of NaKATPase.

Response 10: Thanks for this suggestion. Accordingly, we did the immunocytochemistry to test the level of ATP1A2 protein expression in astrocytes from the cortical tissue of FCD rat following injection of shRNA against ATP1A2. As the **Figure 6I** showed, ATP1A2 protein was seldom expressed in GFAP⁺/shRNA⁺ astrocytes in FCD rat injected with shRNA against ATP1A2, indicating that ATP1A2 expression level was downregulated by shRNA in astrocytes. Thus, we have included this data in the revised manuscript.

Revised Fig. 6I

Comment 11: Authors rationalizes that ChR2 activation in astrocytes causes increase in Na intracellularly and that activates NaKATPase to exchange Na for K intracellularly, which in turn, reduces extracellular K concentration. That is certainly possible, but astrocytes also express other exchangers such as Na/Ca exchanger and Na/H exchanger which could play a role in regulating intracellular Na concentration in response to ChR2 activation. Authors should at least discuss these alternative possibilities.

Response 11: Thanks for this constructive comment. We agree with the reviewer that Na⁺/Ca²⁺ exchanger and Na⁺/H⁺ exchanger may also play a role in regulating intracellular Na⁺ concentration in response to ChR2 activation. Accordingly, we added some discussion in the revised manuscript to bring this point to readers as following:

“Since Na⁺/Ca²⁺ exchanger and Na⁺/H⁺ exchanger would also play an important role in Na⁺ homeostatic (PMID: 26142344), they might also be driven to function in response to the Na⁺ enhancement caused by ChR2 activation. As both Na⁺/Ca²⁺ exchanger and Na⁺/H⁺ exchanger were reported to be involved in seizure modulation (PMID: 19881311, PMID: 26965387), whether they would also contribute to the anti-seizure effect of optogenetic activation of astrocyte need further investigation.”

Comment 12: Although Fig 1i-j shows no generalized seizures (GS) in GFAP-ChR2^{M1} mice (based on analyzing seizure data up to 90 min post-KA), most of these mice do develop GS eventually (about 150 min latency post-KA) as per Extended Data Fig 3. Does this suggest that the effects of optogenetic stimulation of ChR2-expressing astrocytes on KA-induced seizures are transient? Did you obtain EEG data from these mice for a longer term to compare spontaneous seizures?

Response 12: Thanks very much for this interesting question.

In the original manuscript, we continued to record the EEG and observed the behaviors for 120 min (cut off is 120 min). We then compared the latency to GS in *GFAP-ChR2^{M1}* mice and *CAMKII-ARCH^{M1}* mice after photo-stimulation withdrawal. We found that all 4/4 *CAMKII-ARCH^{M1}* mice had GS in ~20 min and 3/4 *GFAP-ChR2^{M1}* mice had GS in 30-60 min and 1/4 *GFAP-ChR2^{M1}* mice did not develop to GS in 120 min (**Extended Data Fig. 3a and 3b**), suggesting anti-seizure effects of optogenetic stimulation of astrocytes might not transient.

To further verify this phenomenon, in the revised manuscript, we added more *GFAP-ChR2^{M1}* mice for post-light recording, among which 5/10 *GFAP-ChR2^{M1}* mice would have GS and 5/10 *GFAP-ChR2^{M1}* mice would still be GS free (**Extended Data Fig. 3c and 3d**). This suggests that the anti-seizure effects of optogenetic stimulation of ChR2-expressing astrocytes on KA-induced seizures are not transient. While for the later epileptogenesis, since the initial severity of SE is quite different between *GFAP-ChR2^{M1}* mice and *CAMKII-ARCH^{M1}* mice, we did not compare the longer-term spontaneous seizures.

Based on this comment, we have added this new data in the revised manuscript as following:

Revised Extended Data Fig. 3

Minor issues:

Comment 13: Fig 2l: Number of neurons in >5 Hz category is too low (n=2). To make any meaningful conclusion, it is desirable to have similar number of neurons (at least 5) recorded under all three firing rate categories. Also, the corresponding text in results section report 4 neurons in >5 Hz category. Please correct it.

Response 13: Thanks for this suggestion. We have recorded more neurons and added sample size in the **Fig. 2k-n** of revised manuscript

Revised Fig. 2k-n:

Comment 14: Extended Data Fig 2e: Which frequency band is compared here?

Response 14: Thanks for this comment. The frequency in Extended Data Fig. 2e is the average frequency of EEG spikes in pilocarpine-induced seizures model. The EEG spikes were counted by Labchart software and averaged in a 10-min-intervals manner. To make it more clearly, we have added more experimental details in the figure legend and revised the title for y axis in Extended Data Figure 2e:

Comment 15: Fig 5h-i: What does explain a huge difference in firing frequency of pyramidal neurons between KA+OUA and KA+shRNA conditions? OUA should inhibit NaKATPase expressed by both astrocytes and neurons, and that makes it inappropriate to test the role of astrocytic NaKATPase. However, its effects are quite comparable to shRNA-mediated knockdown of ATP1A2 in astrocytes.

Response 15: Thank you very much for this question. Accordingly, we have showed the statistical data and found that the firing frequency in KA+OUA was a little higher than that in KA+shRNA, but they did not show the statistical significance (7.706 ± 2.978 Hz vs 6.107 ± 1.097 Hz, $P = 0.5589$, Fig. 5i and Fig. 5k). Interestingly, we can see that there is a big difference in firing frequency of individual neuron in each group. This can be due to the fact that neuronal spiking activity during seizure was highly heterogeneous (PMID: 21441925). As a digitalis drug, ouabain is an inhibitor of the Na^+/K^+ -ATPase and also be used acted as the pharmacological approach to inhibit the astrocytic $\alpha 2$ - Na^+/K^+ -ATPase (PMID: 25344630, PMID: 14730696) due to the factor that $\alpha 2$ isoform is primarily expressed in glial cells in the adult (PMID: 35171651). We agree with the reviewer that ouabain is a non-specific pharmacological inhibition of astrocytic $\alpha 2$ - Na^+/K^+ -ATPase. Then we introduced more specific approach using genetic knockdown of $\alpha 2$ - Na^+/K^+ -ATPase shRNA. Since both ouabain and shRNA could block the astrocytic $\alpha 2$ - Na^+/K^+ -ATPase, their effects are quite comparable, and both could reverse the firing-modulation effect of optogenetic activation of astrocyte. These results indicated that optogenetic activation of astrocyte can suppress neural firing via astrocytic $\alpha 2$ - Na^+/K^+ -ATPase. Thus, we have added some explanation for the difference in firing frequency of individual neuron and included this new data in the revised manuscript.

Fig. 5i and Fig. 5k:

Comment 16: Page 19: “Cannula and electrode implantation surgery” Line 8, Mention Rat before second set of coordinates

Response 16: Thank you for your careful review. This has been corrected as suggested in the revised manuscript.

Comment 17: Page 21: Mention pilocarpine concentration used appropriately.

Response 17: Thank you for this comment. According to the previous study (PMID: 24866701), the concentration of pilocarpine used in our manuscript is 5 M in sterile normal saline and the total injection volume is 200 nL. We corrected the description in the revised manuscript.

Comment 18: Page 12, line 7 (from bottom): in-line citation should be Fig 5l.

Response 18: Thank you for your careful review. This has been corrected in the revised manuscript.

Comment 19: Methods section should include more details regarding EEG, in vivo single-unit recording, and slice electrophysiology.

Response 19: We thank reviewer for this suggestion. We added more details as suggested in the revised manuscript **Methods** section. All the revisions are highlighted by red color.

Reviewer #3 (Remarks to the Author):

Comment 1: *This manuscript reports that selectively activating Channel-rhodopsin (ChR2) in cortical astrocytes attenuates neocortical seizures. The authors show that this attenuation is independent of rapid Ca^{++} signaling, which is confirmed to promote or facilitate seizure genesis consistent previous reports. Instead, the seizure attenuating effect of ChR2-activation of astrocytes is shown to be related to Na^+-K^+ -ATPase-mediate K^+ buffering. The findings both confirm and expand the growing body of information that networks of astrocytes connected via gap junctions play essential roles in buffering extracellular K^+ and that astrocytes can thereby broadly and powerfully influence neuronal excitability.*

From a technical perspective I found the study well designed and well executed, with appropriate controls and sufficient replicates for statistical evaluations. The data are clearly presented, easy to follow, and are convincing. The data do not appear over-interpreted. The findings should be of broad interest regarding astrocyte-neuron interactions in general and adds specific mechanistic information to the long-standing notion that astrocytes are somehow involved in seizures. In particular, the findings point towards the potential for interventions that target astrocyte K^+ buffering to potentially have relevance in the control of intractable seizures. For these reasons I think the study warrants exposure to a broad audience.

Response 1: We appreciate the careful review and enthusiasm expressed by the reviewer. We are also pleased that the reviewer saw the potential for interventions that target astrocytic function have relevance in the control of intractable seizures. Your comments are all valuable and very helpful for revising and improving our paper, as well as the important guiding significance to our research.

Specific comments

Comment 2: *The specificity of the ChR2 etc targeting to astrocytes depends on the specificity of the GFAP-Cre mouse line used. The authors should provide more information in the methods or supplementary data about the mouse line used and the specificity of the targeting to astrocytes using this particular line. Although the authors show reporter targeting data in figure 1, they do not describe the source of the mouse line that they used other than to cite a paper in which they have used the mouse line previously. Going to that cited paper, there is again no specific description of the mouse line other than to refer to a paper from 2006, which states in its title that the mouse line targets GFAP to neuronal progenitors during brain development. At first glance, the use of this mouse seems odd and confusing. The authors should explain in the methods or supplementary data that in adult mice from this line, which is what they used to inject AAV, GFAP-Cre is expressed in the neocortex essentially only in astrocytes and that as their cell counts show, > 98% of neurons did not express reporter.*

Response 2: Thank you very much for raising this important issue and suggestions.

GFAP-Cre mice were generated by Dr. K.D. McCarthy at the University of North Carolina at Chapel Hill. Those mice contains the 2.2-kB fragment of the human GFAP promoter (hGFAP) driving expression of Cre recombinase, surrounded by four copies of genomic insulator. Adult GFAP-Cre mice express Cre recombinase protein in astrocytes but not in neurons or oligodendrocytes (PMID: 16458536, PMID: 28128211).

In our manuscript, we found that ChR2-EYFP virus was restricted to GFAP positive astrocytes and little overlap about ChR2-EYFP virus with NeuN positive neurons ($1.28 \pm 0.25\%$), which is consistent with the previous study

that ChR2-mCherry virus co-localized with the astrocyte-specific marker GFAP, but not with the neuronal marker MAP2 and the NG2-glia marker NG2 in these *GFAP-Cre* mice line (PMID: 28128211). Thus, these data suggesting specificity of the ChR2 targeting to astrocytes.

To make it more clear, we have added more details about GFAP-Cre mouse line in the method “**Animals**” section.

REVIEWERS' COMMENTS

Reviewer #2 (Remarks to the Author):

Authors have conducted additional studies as suggested by the reviewers. Additional data including appropriate control studies have been added, which strengthen the revised manuscript. Although astrocyte-mediated regulation of extracellular potassium concentration is not entirely novel concept (as nicely pointed out by one of the reviewers), this manuscript is a valuable addition to the growing body of work emphasizing astrocyte-mediated regulation of neuronal excitability. I have no further concerns.

Reviewer #3 (Remarks to the Author):

In this revised version, the authors have dealt appropriately with my previous concerns, and as far as I can tell also with the concerns of the other reviewers. Notably, in addition to dealing with technical concerns, the authors have made sincere and successful efforts to improve the readability of the paper. I have no additional comments or concerns and continue to find the paper of high quality and broad interest.

Comments raised by peer reviewers in response to the revised manuscript submission and author responses:

Reviewer #2 (Remarks to the Author):

Comment: Authors have conducted additional studies as suggested by the reviewers. Additional data including appropriate control studies have been added, which strengthen the revised manuscript. Although astrocyte-mediated regulation of extracellular potassium concentration is not entirely novel concept (as nicely pointed out by one of the reviewers), this manuscript is a valuable addition to the growing body of work emphasizing astrocyte-mediated regulation of neuronal excitability. I have no further concerns.

Response: We thank the reviewer for their positive feedback on our revised manuscript.

Reviewer #3 (Remarks to the Author):

Comment: In this revised version, the authors have dealt appropriately with my previous concerns, and as far as I can tell also with the concerns of the other reviewers. Notably, in addition to dealing with technical concerns, the authors have made sincere and successful efforts to improve the readability of the paper. I have no additional comments or concerns and continue to find the paper of high quality and broad interest.

Response: We thank the reviewer for the time on reviewing our revised and manuscript. We are so happy to get the positive feedback from the reviewer.